# If Influence Functions are the Answer, Then What is the Question?

**Juhan Bae**[1,2], **Nathan Ng**[1,2,3], **Alston Lo**[1,2], **Marzyeh Ghassemi**[3], **Roger Grosse**[1,2]
[1]University of Toronto, [2]Vector Institute, [3]Massachusetts Institute of Technology
{jbae, nng, rgrosse}@cs.toronto.edu
alston.lo@mail.utoronto.ca
mghassem@mit.edu

## Abstract

Influence functions efficiently estimate the effect of removing a single training data point on a model's learned parameters. While influence estimates align well with leave-one-out retraining for linear models, recent works have shown this alignment is often poor in neural networks. In this work, we investigate the specific factors that cause this discrepancy by decomposing it into five separate terms. We study the contributions of each term on a variety of architectures and datasets and how they vary with factors such as network width and training time. While practical influence function estimates may be a poor match to leave-one-out retraining for nonlinear networks, we show that they are often a good approximation to a different object we term the *proximal Bregman response function (PBRF)*. Since the PBRF can still be used to answer many of the questions motivating influence functions such as identifying influential or mislabeled examples, our results suggest that current algorithms for influence function estimation give more informative results than previous error analyses would suggest.

## 1 Introduction

The influence function [Hampel, 1974, Cook, 1979] is a classic technique from robust statistics that estimates the effect of deleting a single data example (or a group of data examples) from a training dataset. Formally, given a neural network with learned parameters $\theta^\star$ trained on a dataset $\mathcal{D}$, we are interested in the parameters $\theta^\star_{-\mathbf{z}}$ learned by training on a dataset $\mathcal{D} - \{\mathbf{z}\}$ constructed by deleting a single training example $\mathbf{z}$ from $\mathcal{D}$. By taking the second-order Taylor approximation to the cost function around $\theta^\star$, influence functions approximate the parameters $\theta^\star_{-\mathbf{z}}$ without the computationally prohibitive cost of retraining the model. Since Koh and Liang [2017] first deployed influence functions in machine learning, influence functions have been used to solve various tasks such as explaining model's predictions [Koh and Liang, 2017, Han et al., 2020], relabelling harmful training examples [Kong et al., 2021], carrying out data poisoning attacks [Koh et al., 2022], increasing fairness in models' predictions [Brunet et al., 2019, Schulam and Saria, 2019], and learning data augmentation techniques [Lee et al., 2020].

When the training objective is strongly convex (e.g., as in logistic regression with $L_2$ regularization), influence functions are expected to align well with leave-one-out (LOO) or leave-$k$-out retraining [Koh and Liang, 2017, Koh et al., 2019, Izzo et al., 2021]. However, Basu et al. [2020a] showed that influence functions in neural networks often do not accurately predict the effect of retraining the model and concluded that influence estimates are often "fragile" and "erroneous". Because of the poor match between influence estimates and LOO retraining, influence function methods are often evaluated with alternative metrics such as the detection rate of maliciously corrupted examples using influence scores [Khanna et al., 2019, Koh and Liang, 2017, Schioppa et al., 2021, K and Søgaard,

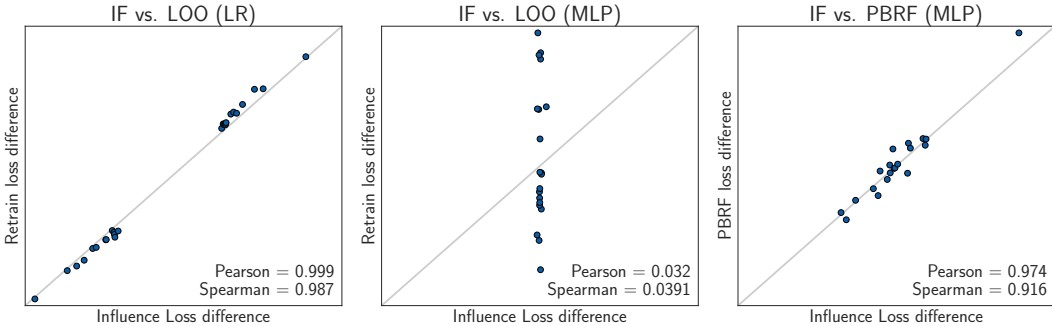

**Figure 1:** Comparison of test loss differences computed by influence function (IF), leave-one-out (LOO) retraining, and our proximal Bregman response function (PBRF). Each point corresponds to the individual effect of removing one training example. Influence estimates align well with true retraining for (**left**) logistic regression (LR) but poorly for (**middle**) multilayer perceptrons (MLP). While influence functions in neural networks do not accurately predict the effect of retraining the model, they are still a good approximation to (**right**) the PBRF.

2021]. However, these indirect signals make it difficult to develop algorithmic improvements to influence function estimation. If one is interested in improving certain aspects of influence function estimation, such as the linear system solver, it would be preferable to have a well-defined quantity that influence function estimators are approximating so that algorithmic choices could be directly evaluated based on the accuracy of their estimates.

In this work, we investigate the source of the discrepancy between influence functions and LOO retraining in neural networks. We decompose the discrepancy into five components: (1) the difference between cold-start and warm-start response functions (a concept elaborated on below), (2) an implicit proximity regularizer, (3) influence estimation on non-converged parameters, (4) linearization, and (5) approximate solution of a linear system. This decomposition was chosen to capture all gaps and errors caused by approximations and assumptions made in applying influence functions to neural networks. We empirically evaluate the contributions of each component on binary classification, regression, image reconstruction, image classification, and language modeling tasks and show that, across all tasks, components (1–3) are most responsible for the discrepancy between influence functions and LOO retraining. We further investigate how the contribution of each component changes in response to the change in network width and depth, weight decay, training time, damping, and the number of data points being removed.

Moreover, we show that while influence functions for neural networks are often a poor match to LOO retraining, they are a much better match to what we term the *proximal Bregman response function (PBRF)*. Intuitively, the PBRF approximates the effect of removing a data point while trying to keep the predictions consistent with those of the (partially) trained model. From this perspective, we reframe misalignment components (1–3) as simply reflecting the difference between LOO retraining and the PBRF. The gap between the influence function estimate and the PRBF only comes from sources (4) and (5), which we found empirically to be at least an order of magnitude smaller for most neural networks. As a result, on a wide variety of tasks, influence functions closely align with the PBRF while failing to approximate the effect of retraining the model, as shown in Figure 1.

The PBRF can be used for many of the same use cases that have motivated influence functions, such as finding influential or mislabeled examples [Schioppa et al., 2021] and carrying out data poisoning attacks [Koh and Liang, 2017, Koh et al., 2022], and can therefore be considered an alternative to LOO retraining as a gold standard for evaluating influence functions. Hence, we conclude that influence functions applied to neural networks are not inherently "fragile" as is often believed [Basu et al., 2020a], but instead can be seen as giving accurate answers to a different question than is normally assumed.

## 2   Related Work

Instance-based interpretability methods are a class of techniques that explain a model's predictions in terms of the examples on which the model was trained. Methods of this type include TracIn [Pruthi et al., 2020], Representer Point Selection [Yeh et al., 2018], Grad-Cos and Grad-Dot [Charpiat et al., 2019, Hanawa et al., 2021], MMD-critic [Kim et al., 2016], unconditional counterfactual

explanations [Wachter et al., 2018], and of central focus in this paper, influence functions. Since its adoption in machine learning by Koh and Liang [2017], multiple extensions and improvements upon influence functions have also been proposed, such as variants that use Fisher kernels [Khanna et al., 2019], higher-order approximations [Basu et al., 2020b], tricks for faster and scalable inference [Guo et al., 2021, Schioppa et al., 2021], group influence formulations [Koh et al., 2019, Basu et al., 2020b], and relative local weighting [Barshan et al., 2020]. However, many of these methods rely on the same strong assumptions made in the original influence function derivation that the objective needs to be strongly convex and influence functions must be computed on the optimal parameters.

In general, influence functions are assumed to approximate the effects of leave-one-out (LOO) retraining from scratch, the parameters of the network that are trained without a data point of interest. Hence, measuring the quality of influence functions is often performed by analyzing the correlation between LOO retraining and influence function estimations [Koh and Liang, 2017, Basu et al., 2020a,b, Yang and Chaudhuri, 2022]. However, recent empirical analyses have demonstrated the fragility of influence functions and a fundamental misalignment between their assumed and actual effects [Basu et al., 2020a, Ghorbani et al., 2019, K and Søgaard, 2021]. For example, Basu et al. [2020a] argued that the accuracy of influence functions in deep networks is highly sensitive to network width and depth, weight decay strength, inverse-Hessian vector product estimation methodology, and test query point by measuring the alignment between influence functions and LOO retraining. Because of the inherent misalignment between influence estimations and LOO retraining in neural networks, many works often evaluate the accuracy of the influence functions on an alternative metric, such as the recovery rate of maliciously mislabelled or poisoned data using influence functions [Khanna et al., 2019, Koh and Liang, 2017, Schioppa et al., 2021, K and Søgaard, 2021]. In this work, instead of interpreting the misalignment between influence functions and LOO retraining as a failure, we claim that it simply reflects that influence functions answer a different question than is typically assumed.

## 3 Background

Consider a prediction task from an input space $\mathcal{X}$ to a target space $\mathcal{T}$ where we are given a finite training dataset $\mathcal{D}_{\text{train}} = \{(\mathbf{x}^{(i)}, \mathbf{t}^{(i)})\}_{i=1}^{N}$. Given a data point $\mathbf{z} = (\mathbf{x}, \mathbf{t})$, let $\mathbf{y} = f(\boldsymbol{\theta}, \mathbf{x})$ be the prediction of the network parameterized by $\boldsymbol{\theta} \in \mathbb{R}^d$ and $\mathcal{L}(\mathbf{y}, \mathbf{t})$ be the loss (e.g., squared error or cross-entropy). We aim to solve the following optimization problem:

$$\boldsymbol{\theta}^{\star} = \underset{\boldsymbol{\theta} \in \mathbb{R}^d}{\arg\min} \, \mathcal{J}(\boldsymbol{\theta}) = \underset{\boldsymbol{\theta} \in \mathbb{R}^d}{\arg\min} \, \frac{1}{N} \sum_{i=1}^{N} \mathcal{L}(f(\boldsymbol{\theta}, \mathbf{x}^{(i)}), \mathbf{t}^{(i)}), \tag{1}$$

where $\mathcal{J}(\cdot)$ is the cost function. If the regularization (e.g., $L_2$ regularization) is imposed in the cost function, we fold the regularization terms into the loss function. We summarize the notation used in this paper in Appendix A.

### 3.1 Downweighting a Training Example

The training objective in Eqn. 1 aims to find the parameters that minimize the average loss on all training examples. Herein, we are interested in studying the change in optimal model parameters when a particular training example $\mathbf{z} = (\mathbf{x}, \mathbf{t}) \in \mathcal{D}_{\text{train}}$ is removed from the training dataset, or more generally, when the data point $\mathbf{z}$ is downweighted by an amount $\epsilon \in \mathbb{R}$. Formally, this corresponds to minimizing the following downweighted objective:

$$\boldsymbol{\theta}^{\star}_{-\mathbf{z}, \epsilon} = \underset{\boldsymbol{\theta} \in \mathbb{R}^d}{\arg\min} \, \mathcal{Q}_{-\mathbf{z}}(\boldsymbol{\theta}, \epsilon) = \underset{\boldsymbol{\theta} \in \mathbb{R}^d}{\arg\min} \, \mathcal{J}(\boldsymbol{\theta}) - \mathcal{L}(f(\boldsymbol{\theta}, \mathbf{x}), \mathbf{t})\epsilon. \tag{2}$$

When $\epsilon = 1/N$, the downweighted objective reduces to the cost over the dataset with the example $\mathbf{z}$ removed, up to a constant factor. To see how the optimum of the downweighted objective responds to changes in the downweighting factor $\epsilon$, we define the *response function* $r^{\star}_{-\mathbf{z}} \colon \mathbb{R} \to \mathbb{R}^d$ by:

$$r^{\star}_{-\mathbf{z}}(\epsilon) = \underset{\boldsymbol{\theta} \in \mathbb{R}^d}{\arg\min} \, \mathcal{Q}_{-\mathbf{z}}(\boldsymbol{\theta}, \epsilon), \tag{3}$$

where we assume that the downweighted objective is strongly convex and hence the solution to the downweighted objective is unique given some factor $\epsilon$. Under these assumptions, note that $r^{\star}_{-\mathbf{z}}(0) = \boldsymbol{\theta}^{\star}$ and the response function is differentiable at 0 by the Implicit Function Theorem [Krantz

and Parks, 2002, Griewank and Walther, 2008]. Influence functions approximate the response function by performing a first-order Taylor expansion around $\epsilon_0 = 0$:

$$r^{\star}_{-\mathbf{z},\text{lin}}(\epsilon) = r^{\star}_{-\mathbf{z}}(\epsilon_0) + \frac{\mathrm{d}r^{\star}_{-\mathbf{z}}}{\mathrm{d}\epsilon}\bigg|_{\epsilon=\epsilon_0}(\epsilon - \epsilon_0) = \boldsymbol{\theta}^{\star} + (\nabla^2_{\boldsymbol{\theta}}\mathcal{J}(\boldsymbol{\theta}^{\star}))^{-1}\nabla_{\boldsymbol{\theta}}\mathcal{L}(f(\boldsymbol{\theta}^{\star}, \mathbf{x}), \mathbf{t})\epsilon. \quad (4)$$

We refers readers to Van der Vaart [2000] and Appendix B for a detailed derivation. The optimal parameters trained without $\mathbf{z}$ can then be approximated by plugging in $\epsilon = 1/N$ to Eqn. 4.

Influence functions can further approximate the loss of a particular test point $\mathbf{z}_{\text{test}} = (\mathbf{x}_{\text{test}}, \mathbf{t}_{\text{test}})$ when a data point $\mathbf{z}$ is eliminated from the training set using the chain rule [Koh and Liang, 2017]:

$$\begin{aligned}
&\mathcal{L}(f(r^{\star}_{-\mathbf{z},\text{lin}}(1/N), \mathbf{x}_{\text{test}}), \mathbf{t}_{\text{test}}) \\
&\approx \mathcal{L}(f(\boldsymbol{\theta}^{\star}, \mathbf{x}_{\text{test}}), \mathbf{t}_{\text{test}}) + \frac{1}{N}\nabla_{\boldsymbol{\theta}}\mathcal{L}(f(\boldsymbol{\theta}^{\star}, \mathbf{x}_{\text{test}}), \mathbf{t}_{\text{test}})^{\top}\frac{\mathrm{d}r^{\star}_{\mathbf{z}}}{\mathrm{d}\epsilon}\bigg|_{\epsilon=0} \\
&= \mathcal{L}(f(\boldsymbol{\theta}^{\star}, \mathbf{x}_{\text{test}}), \mathbf{t}_{\text{test}}) + \frac{1}{N}\nabla_{\boldsymbol{\theta}}\mathcal{L}(f(\boldsymbol{\theta}^{\star}, \mathbf{x}_{\text{test}}), \mathbf{t}_{\text{test}})^{\top}(\nabla^2_{\boldsymbol{\theta}}\mathcal{J}(\boldsymbol{\theta}^{\star}))^{-1}\nabla_{\boldsymbol{\theta}}\mathcal{L}(f(\boldsymbol{\theta}^{\star}, \mathbf{x}), \mathbf{t}).
\end{aligned} \quad (5)$$

### 3.2 Influence Function Estimation in Neural Networks

Influence functions face two main challenges when deployed on neural networks. First, the influence estimation (shown in Eqn. 4) requires computing an inverse Hessian-vector product (`iHVP`). Unfortunately, storing and inverting the Hessian requires $O(d^3)$ operations and is infeasible to compute for modern neural networks. Instead, Koh and Liang [2017] tractably approximate the `iHVP` using truncated non-linear conjugate gradient (`CG`) [Martens et al., 2010] or the `LiSSA` algorithm [Agarwal et al., 2016]. Both approaches avoid explicit computation of the Hessian inverse (see Appendix G for details) and only require $O(Nd)$ operations to approximate the influence function.

Second, the derivation of influence functions assumes a strongly convex objective, which is often not satisfied for neural networks. The Hessian may be singular, especially when the parameters have not fully converged, due to non-positive eigenvalues. To enforce positive-definiteness of the Hessian, Koh and Liang [2017] add a damping term in the `iHVP`. Teso et al. [2021] further approximate the Hessian with the Fisher information matrix (which is equivalent to the Gauss-Newton Hessian [Martens, 2014] for commonly used loss functions such as cross-entropy) as follows:

$$r^{\star}_{-\mathbf{z},\text{damp,lin}}(\epsilon) \approx \boldsymbol{\theta}^{\star} + (\mathbf{J}^{\top}_{\mathbf{y}\boldsymbol{\theta}^{\star}}\mathbf{H}_{\mathbf{y}^{\star}}\mathbf{J}_{\mathbf{y}\boldsymbol{\theta}^{\star}} + \lambda\mathbf{I})^{-1}\nabla_{\boldsymbol{\theta}}\mathcal{L}(f(\boldsymbol{\theta}^{\star}, \mathbf{x}), \mathbf{t})\epsilon, \quad (6)$$

where $\mathbf{J}_{\mathbf{y}\boldsymbol{\theta}^{\star}}$ is the parameter-output Jacobian and $\mathbf{H}_{\mathbf{y}^{\star}}$ is the Hessian of the cost with respect to the network outputs both evaluated on the optimal parameters $\boldsymbol{\theta}^{\star}$. Here, $\mathbf{G}^{\star} = \mathbf{J}^{\top}_{\mathbf{y}\boldsymbol{\theta}^{\star}}\mathbf{H}_{\mathbf{y}^{\star}}\mathbf{J}_{\mathbf{y}\boldsymbol{\theta}^{\star}}$ is the Gauss-Newton Hessian (GNH) and $\lambda > 0$ is a damping term to ensure the invertibility of GNH. Unlike the Hessian, the GNH is guaranteed to be positive semidefinite as long as the loss function is convex as a function of the network outputs [Martens et al., 2010].

## 4 Understanding the Discrepancy between Influence Function and LOO Retraining in Neural Networks

In this section, we investigate several factors responsible for the misalignment between influence functions and LOO retraining. Specifically, we decompose the misalignment into five separate terms: (1) the warm-start gap, (2) the damping gap, (3) the non-convergence gap, (4) the linearization error, and (5) the solver error. This decomposition captures all approximations and assumption violations when deploying influence functions in neural networks. By summing the parameter (or outputs) differences introduced by each term we can bound the parameter (or outputs) difference between LOO retraining and influence estimates. We use the term "gap" rather than "error" for the first three terms to emphasize that they reflect differences between solutions to different influence-related questions, rather than actual errors.

For all models we investigate, we find that the first three sources dominate the misalignment, indicating that the misalignment reflects not algorithmic errors but rather the fact that influence function estimators are answering a different question from what is normally assumed. All proximal objectives are summarized in Table 1 and we provide the derivations in Appendix B.

## 4.1 Warm-Start Gap: Non-Strongly Convex Training Objective

By taking a first-order Taylor approximation of the response function at $\epsilon_0 = 0$ (Eqn. 4), influence functions approximate the effect of removing a data point $\mathbf{z}$ at a local neighborhood of the optimum $\boldsymbol{\theta}^\star$. Hence, influence approximation has a more natural connection to the retraining scheme that initializes the network at the current optimum $\boldsymbol{\theta}^\star$ (*warm-start retraining*) than the scheme that initializes the network randomly (*cold-start retraining*). The warm-start optimum is equivalent to the cold-start optimum when the objective is strongly convex (where the solution to the response function is unique), making the influence estimation close to the LOO retraining on logistic regression with $L_2$ regularization.

However, the equivalence between warm-start and cold-start optima is not typically guaranteed in neural networks [Vicol et al., 2022a]. Particularly, in the over-parametrized regime ($N < d$), neural networks exhibit multiple global optima, and their converged solutions depend highly on the specifics of the optimization dynamics [Lee et al., 2019, Arora et al., 2019, Bartlett et al., 2020, Amari et al., 2020]. For quadratic cost functions, gradient descent with initialization $\boldsymbol{\theta}^0$ converges to the

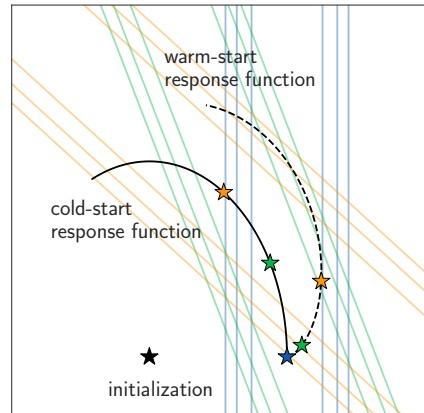

**Figure 2:** Cold-start (initialized from black star) and warm-start (initialized from blue star) response functions for quadratic cost function. Each contour represents the cost function at some $\epsilon$. Because gradient descent converges to a minimum-norm solution, the warm-start and cold-start optima are not equivalent.

optimum that achieves the minimum $L_2$ distance from $\boldsymbol{\theta}^0$ [Hastie et al., 2022]. This phenomenon of the converged parameters being dependent on the initialization hinders influence functions from accurately predicting the effect of retraining the model from scratch as shown in Figure 2. We denote the discrepancy between cold-start and warm-start optima as **warm-start gap**.

## 4.2 Proximity Gap: Addition of Damping Term in `iHVP`

In practical settings, we often impose a damping term (Eqn. 6) in influence approximations to ensure that the cost Hessian is positive-definite and hence invertible. As adding a damping term in influence estimations is equivalent to adding $L_2$ regularization to the cost function [Martens et al., 2010], when damping is used, influence functions can be seen as linearizing the following *proximal response function* at $\epsilon_0 = 0$:

$$r^\star_{-\mathbf{z},\text{damp}}(\epsilon) = \underset{\boldsymbol{\theta} \in \mathbb{R}^d}{\arg\min} \, \mathcal{Q}_{-\mathbf{z}}(\boldsymbol{\theta}, \epsilon) + \frac{\lambda}{2}\|\boldsymbol{\theta} - \boldsymbol{\theta}^\star\|^2. \tag{7}$$

See Appendix B.2 for the derivation. Note that $\lambda > 0$ is a damping strength and our use of "proximal" is based on the notion of proximal equilibria [Farnia and Ozdaglar, 2020]. Intuitively, the proximal objective in Eqn. 7 not only minimizes the downweighted objective but also encourages the parameters to stay close to the optimal parameters at $\epsilon_0 = 0$. Hence, when the damping term is used in the `iHVP`, influence functions aim at approximating the warm-start retraining scheme with a proximity term that penalizes the $L_2$ distance between the new estimate and the optimal parameters. We call the discrepancy between the warm-start and proximal warm-start optima the **proximity gap**.

Interestingly, past works have observed that for quadratic cost functions, early stopping has a similar effect to $L_2$ regularization [Vicol et al., 2022a, Ali et al., 2019]. Therefore, the proximal response function can be thought of as capturing how gradient descent will respond to a dataset perturbation if it takes only a limited number of steps starting from the warm-start solution.

## 4.3 Non-Convergence Gap: Influence Estimation on Non-Converged Parameters

Thus far, our analysis has assumed that influence functions are computed on fully converged parameters $\boldsymbol{\theta}^\star$ at which the gradient of the cost is $\mathbf{0}$. However, in neural network training, we often terminate the optimization procedure before reaching the exact optimum due to several reasons, including having limited computational resources or to avoid overfitting [Bengio, 2012]. In such situations, much of the change in the parameters from LOO retraining simply reflects the effect of training for

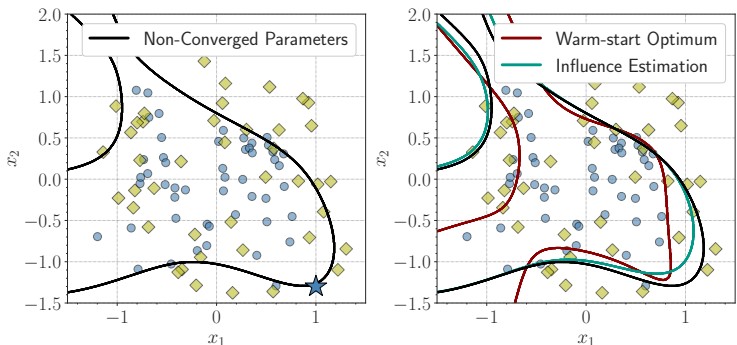

**Figure 3:** Decision boundaries for a partially trained binary classifier. We consider removing a data point located at right-bottom corner denoted as $\star$. While the influence estimation makes a local change on the data point of interest, the (warm-start) LOO retraining globally updates the parameters to better fit other data points (a nuisance from the perspective of understanding influence).

| Error | Objective | Init |
|---|:---:|:---:|
| Cold-start | $\mathcal{J}(\boldsymbol{\theta}) - \mathcal{L}(f(\boldsymbol{\theta}, \mathbf{x}), \mathbf{t})\epsilon$ | $\boldsymbol{\theta}^0$ |
| + Warm-start | $\mathcal{J}(\boldsymbol{\theta}) - \mathcal{L}(f(\boldsymbol{\theta}, \mathbf{x}), \mathbf{t})\epsilon$ | $\boldsymbol{\theta}^s$ |
| + Proximity | $\mathcal{J}(\boldsymbol{\theta}) - \mathcal{L}(f(\boldsymbol{\theta}, \mathbf{x}), \mathbf{t})\epsilon + \frac{\lambda}{2}\|\boldsymbol{\theta} - \boldsymbol{\theta}^s\|^2$ | $\boldsymbol{\theta}^s$ |
| + Non-Convergence | $\frac{1}{N}\sum_{i=1}^{N} D_{\mathcal{L}^{(i)}}(f(\boldsymbol{\theta}, \mathbf{x}^{(i)}), f(\boldsymbol{\theta}^s, \mathbf{x}^{(i)})) - \mathcal{L}(f(\boldsymbol{\theta}, \mathbf{x}), \mathbf{t})\epsilon + \frac{\lambda}{2}\|\boldsymbol{\theta} - \boldsymbol{\theta}^s\|^2$ | $\boldsymbol{\theta}^s$ |
| + Linearization | $\frac{1}{N}\sum_{i=1}^{N} D_{\mathcal{L}^{(i)}_{\mathrm{quad}}}(f_{\mathrm{lin}}(\boldsymbol{\theta}, \mathbf{x}^{(i)}), f(\boldsymbol{\theta}^s, \mathbf{x}^{(i)})) - \nabla_{\boldsymbol{\theta}}\mathcal{L}(f(\boldsymbol{\theta}^s, \mathbf{x}), \mathbf{t})^\top \boldsymbol{\theta}\epsilon + \frac{\lambda}{2}\|\boldsymbol{\theta} - \boldsymbol{\theta}^s\|^2$ | $\boldsymbol{\theta}^s$ |

**Table 1:** Summary of proximal objectives that influence functions aim to approximate when the network is non-strongly convex, a damping term is used, and influence functions are computed on non-converged parameters. The final linearization reflects the second-order approximation that influence functions utilize.

longer, rather than the effect of removing a training example, as illustrated in Figure 3. What we desire from influence functions is to understand the effect of removing the training example; the effect of extended training is simply a nuisance. Therefore, to the extent that this factor contributes to the misalignment between influence functions and LOO retraining, influence functions are arguably *more useful* than LOO retraining.

Since training the network to convergence may be impractical or undesirable, we instead modify the response function by replacing the original training objective with a similar one *for which the (possibly non-converged) final parameters $\boldsymbol{\theta}^s$ are optimal*. Here, we assume the loss function $\mathcal{L}(\cdot, \cdot)$ is convex as a function of the network outputs; this is true for commonly used loss functions such as squared error or cross-entropy. We replace the training loss with a term that penalizes mismatch to the predictions made by $\boldsymbol{\theta}^s$ (hence implying that $\boldsymbol{\theta}^s$ is optimal). Our *proximal Bregman response function (PBRF)* is defined as follows:

$$r^b_{-\mathbf{z},\mathrm{damp}}(\epsilon) = \arg\min_{\boldsymbol{\theta} \in \mathbb{R}^d} \frac{1}{N} \sum_{i=1}^{N} D_{\mathcal{L}^{(i)}}(f(\boldsymbol{\theta}, \mathbf{x}^{(i)}), f(\boldsymbol{\theta}^s, \mathbf{x}^{(i)})) - \mathcal{L}(f(\boldsymbol{\theta}, \mathbf{x}), \mathbf{t})\epsilon + \frac{\lambda}{2}\|\boldsymbol{\theta} - \boldsymbol{\theta}^s\|^2,$$

(8)

where $D_{\mathcal{L}^{(i)}}(\cdot, \cdot)$ is the Bregman divergence defined as:

$$D_{\mathcal{L}^{(i)}}(\mathbf{y}, \mathbf{y}^s) = \mathcal{L}(\mathbf{y}, \mathbf{t}^{(i)}) - \mathcal{L}(\mathbf{y}^s, \mathbf{t}^{(i)}) - \nabla_{\mathbf{y}}\mathcal{L}(\mathbf{y}^s, \mathbf{t}^{(i)})^\top(\mathbf{y} - \mathbf{y}^s).$$

(9)

The PBRF defined in Eqn. 8 is composed of three terms. The first term measures the functional discrepancy between the current estimate and the parameters $\boldsymbol{\theta}^s$ in Bregman divergence, and its role is to prevent the new estimate from drastically altering the predictions on the training dataset. One way of understanding this term in the cases of squared error or cross-entropy losses is that it is equivalent to the training error on a dataset where the original training labels are replaced with soft targets obtained from the predictions made by $\boldsymbol{\theta}^s$. The second term is the negative loss on the data point $\mathbf{z} = (\mathbf{x}, \mathbf{t})$, which aims to respond to the deletion of a training example. The final term is simply the proximity term described before. In Appendix B.3, we further show that the influence

function on non-converged parameters is equivalent to the first-order approximation of PBRF instead of the first-order approximation of proximal response function for linear models.

Rather than computing the LOO retrained parameters by performing $K$ additional optimization steps under the original training objective, we can instead perform $K$ optimization steps under the proximal Bregman objective. The difference between the resulting parameter vectors is what we call the **non-convergence gap**.

### 4.4 Linearization Error: A First-order Taylor Approximation of the Response Function

The key idea behind influence functions is the linearization of the response function. To simulate the local approximations made in influence functions, we define the linearized PBRF as:

$$
\begin{aligned}
r^b_{-\mathbf{z},\text{damp},\text{lin}}(\epsilon) = \arg\min_{\boldsymbol{\theta} \in \mathbb{R}^d} \frac{1}{N} \sum_{i=1}^{N} D_{\mathcal{L}_{\text{quad}}^{(i)}}\left(f_{\text{lin}}(\boldsymbol{\theta}, \mathbf{x}^{(i)}), f(\boldsymbol{\theta}^s, \mathbf{x}^{(i)})\right) \\
- \nabla_{\boldsymbol{\theta}} \mathcal{L}(f(\boldsymbol{\theta}^s, \mathbf{x}), \mathbf{t})^\top \boldsymbol{\theta} \epsilon + \frac{\lambda}{2} \|\boldsymbol{\theta} - \boldsymbol{\theta}^s\|^2,
\end{aligned}
\tag{10}
$$

where $\mathcal{L}_{\text{quad}}(\cdot, \cdot)$ is the second-order expansion of the loss around $\mathbf{y}^s$ and $f_{\text{lin}}(\cdot, \cdot)$ is the linearization of the network outputs with respect to the parameters. The optimal solution to the linearized PBRF is equivalent to the influence estimation at the parameters $\boldsymbol{\theta}^s$ with the GNH approximation and a damping term $\lambda$ (see Appendix B.4 for the derivation).

As the linearized PBRF relies on several local approximations, the linearization error increases when the downweighting factor magnitude $|\epsilon|$ is large or the PBRF is highly non-linear. We refer to the discrepancy between the PBRF and linearized PBRF as the **linearization error**.

### 4.5 Solver Error: A Crude Approximation of `iHVP`

As the precise computation of the `iHVP` is computationally infeasible, in practice, we use truncated `CG` or `LiSSA` to efficiently approximate influence functions [Koh and Liang, 2017]. Unfortunately, these efficient linear solvers introduce additional error by crudely approximating the `iHVP`. Moreover, different linear solvers can introduce specific biases in the influence estimation. For example, Vicol et al. [2022b] show that the truncated `LiSSA` algorithm implicitly adds an additional damping term in the `iHVP`. We use **solver error** to refer to the difference between the linearized PBRF and the influence estimation computed by a linear solver.

Interestingly, Koh and Liang [2017] reported that the `LiSSA` algorithm gave more accurate results than `CG`. We have determined that this difference resulted not from any inherent algorithmic advantage to `LiSSA`, but rather from the fact that the software used different damping strengths for the two algorithms, thereby resulting in different weightings of the proximity term in the proximal response function.

## 5 PBRF: The Question Influence Functions are Really Answering

The PBRF (Eqn. 8) approximates the effect of removing a data point while trying to keep the predictions consistent with those of the (partially) trained model. Since the discrepancy between the PBRF and influence function estimates is only due to the linearization and solver errors, the PBRF can be thought of as better representing the question that influence functions are trying to answer. Reframing influence functions in this way means that the PBRF can be regarded as a gold-standard ground truth for evaluating methods for influence function approximation. Existing analyses of influence functions [Basu et al., 2020a] rely on generating LOO retraining ground truth estimates by imposing strong $L_2$ regularization or training till convergence without early stopping. However, these conditions do not accurately reflect the typical way neural networks are trained in practice. In contrast, our PBRF formulation does not require the addition of any regularizers or modified training regimes and can be easily optimized.

In addition, although the PBRF may not necessarily align with LOO retraining due to the warm-start, proximity, and non-convergence gaps, the motivating use cases for influence functions typically do not rely on exact LOO retraining. This means that the PBRF can be used in place of LOO retraining for

many tasks such as identifying influential or mislabelled examples, as demonstrated in Appendix D.3. In these cases, influence functions are still useful since they provide an efficient way of approximating PBRF estimates.

# 6 Experiments

Our experiments investigate the following questions: (1) What factors discussed in Section 4 contribute most to the misalignment between influence functions and LOO retraining? (2) While influence functions fail to approximate the effect of retraining, do they accurately approximate the PBRF? (3) How do changes in weight decay, damping, the number of total epochs, and the number of removed training examples affect each source of misalignment?

In all experiments, we first train the base network with the entire dataset to obtain the parameters $\boldsymbol{\theta}^s$. We repeat the training procedure 20 times with a different random training example deleted. The cold-start retraining begins from the same initialization used to train $\boldsymbol{\theta}^s$. All proximal objectives are trained with initialization $\boldsymbol{\theta}^s$ for 50% of the epochs used to train the base network. Lastly, we use the `LiSSA` algorithm with GNH approximation to compute influence functions.

Since we are primarily interested in the effect of deleting a data point on model's predictions, we measure the discrepancy of each gap and error using the average $L_2$ distance between networks' outputs $\mathbb{E}_{(\mathbf{x}, \cdot) \sim \mathcal{D}_{\text{train}}}[\| f(\boldsymbol{\theta}, \mathbf{x}) - f(\boldsymbol{\theta}', \mathbf{x}) \|]$ on the training dataset. We provide the full experimental set-up and additional experiments in Appendix C and D, respectively.

## 6.1 Influence Misalignment Decomposition

We first applied our decomposition to various models trained on a broad range of tasks covering binary classification, regression, image reconstruction, image classification, and language modeling. The summary of our results is provided in Figure 4 and Table 5 (Appendix E). Across all tasks, we found that the first three sources dominate the misalignment, indicating influence function estimators are answering a different question from what is normally assumed. Small linearization and solver errors indicate that influence functions accurately answer the modified question (PBRF).

**Logistic Regression.** We analyzed the logistic regression (LR) model trained on the Cancer and Diabetes classification datasets from the UCI collection [Dua and Graff, 2017]. We trained the model using `L-BFGS` [Liu and Nocedal, 1989] with $L_2$ regularization of 0.01 and damping term of $\lambda = 0.001$. As the training objective is strongly convex and the base model parameters were trained till convergence, in Table 5, we observed that each source of misalignment is significantly low. Hence, in the case of logistic regression with $L_2$ regularization, influence functions accurately capture the effect of retraining the model without a data point.

**Multilayer Perceptron.** Next, we applied our analysis to the 2-hidden layer Multilayer Perceptron (MLP) with ReLU activations. We conducted the experiments in two settings: (1) regression on the Concrete and Energy datasets from the UCI collection and (2) image classification on 10% of the MNIST [Deng, 2012] and FashionMNIST [Xiao et al., 2017] datasets, following the set-up from Koh and Liang [2017] and Basu et al. [2020a]. We trained the networks for 1000 epochs using stochastic gradient descent (SGD) with a batch size of 128 and set a damping strength of $\lambda = 0.001$.

As opposed to linear models, MLPs violate the assumptions in the influence derivation and we observed an increase in gaps and errors on all five factors. We observed that warm-start, proximity, and the non-convergence gaps contribute more to the misalignment than linearization and solver errors. The average network's predictions for PBRF were similar to that computed by the `LiSSA` algorithm, demonstrating that influence functions are still a good approximation to PBRF.

**Autoencoder.** Next, we applied our framework to an 8-layer autoencoder (AE) on the full MNIST dataset. We followed the experimental set-up from Martens and Grosse [2015], where the encoder and decoder each consist of 4 fully-connected layers with sigmoid activation functions. We trained the network for 1000 epochs using SGD with momentum. We set the batch size to 1024, used $L_2$ regularization of $10^{-5}$ with a damping factor of $\lambda = 0.001$. In accordance with the findings from our MLP experiments, the warm-start, proximity, and non-convergence gaps were more significant than the linearization and solver errors, and influence functions accurately predicted the PBRF.

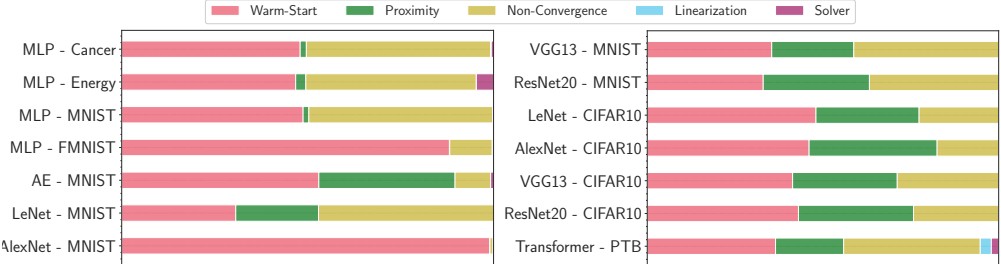

**Figure 4:** Decomposition of the discrepancy between influence functions and LOO retraining into (1) warm-start gap, (2) proximal gap, (3) non-convergence gap, (4) linearization error, and (5) solver error for each model and dataset. The size of each component is measured by the $L_2$ distance between the networks' outputs on the training dataset.

**Convolutional Neural Networks.** To investigate the source of discrepancy on larger-scale networks, we trained a set of convolutional neural networks of increasing complexity and size. Namely, LeNet [Lecun et al., 1998], AlexNet [Krizhevsky et al., 2012], VGG13 Simonyan and Zisserman [2014], and ResNet-20 [He et al., 2015] were trained on 10% of the MNIST dataset and the full CIFAR10 [Krizhevsky, 2009] dataset. We trained the base network for 200 epochs on both datasets with a batch size of 128. For MNIST, we kept the learning rate fixed throughout training, while

| Model | Cold-Start | | Warm-Start | | PBRF | |
|---|---|---|---|---|---|---|
| | P | S | P | S | P | S |
| MLP | -0.55 | 0.01 | 0.22 | 0.35 | **0.98** | **0.99** |
| LeNet | -0.19 | 0.12 | 0.32 | 0.25 | **0.93** | **0.52** |
| AlexNet | -0.16 | -0.08 | 0.51 | 0.58 | **0.99** | **0.99** |
| VGG13 | 0.45 | -0.07 | -0.28 | -0.51 | **0.98** | **0.77** |
| ResNet-20 | 0.09 | -0.06 | 0.02 | 0.09 | **0.81** | **0.76** |

**Table 2:** Comparison of test loss differences computed by influence function, cold-start retraining, warm-start retraining, and PBRF on MNIST dataset. We show Pearson (P) and Spearman rank-order (S) correlation when compared to influence estimates.

for CIFAR10, we decayed the learning rate by a factor of 5 at epochs 60, 120, and 160, following Zagoruyko and Komodakis [2016]. We used $L_2$ regularization with strength $5 \cdot 10^{-4}$ and a damping factor of $\lambda = 0.001$. Consistent with the findings from our MLP and autoencoder experiments, the first three gaps were more significant than linearization and solver errors.

We further compared influence functions' approximations on the difference in test loss when a random training data point is removed with the value obtained from cold-start retraining, warm-start retraining, and PBRF in Table 2. We used both Pearson [Sedgwick, 2012] and Spearman rank-order correlation [Spearman, 1961] to measure the alignment. While the test loss predicted by influence functions does not align well with the values obtained by cold-start and warm-start retraining schemes, they show high correlations when compared to the estimates given by PBRF.

**Transformer.** Finally, we trained 2-layer Transformer language models on the Penn Treebank (PTB) [Marcus et al., 1993] dataset. We set the number of hidden dimensions to 256 and the number of attention heads to 2. As we observed that model overfits after a few epochs of training, we trained the base network for 10 epochs using Adam. Notably, we observed that the non-convergence gap had the most considerable contribution to the discrepancy between influence functions and LOO retraining. Consistent with our previous findings, the first tree gaps had more impact on the discrepancy compared to linearization and solver errors.

### 6.2 Factors in Influence Misalignment

We further analyzed how the contribution of each component changes in response to changes in network width and depth, training time, weight decay, damping, and the percentage of data removed. We used an MLP trained on 10% of the MNIST dataset and summarized results in Figure 5.

**Width and Depth.** As we increase the width of the network, we observe a decrease in the linearization error. This is consistent with previous observations that networks behave more linearly as the width is increased [Lee et al., 2019]. In contrast to the findings from Basu et al. [2020a], we did not observe a strong relationship between the contribution of the components and the depth of the network.

**Training Time.** Unsurprisingly, as we increase the number of training epochs, we observe a decrease in the non-convergence gap. We hypothesize that, as we increase the training epoch, the cost gradient reaches **0**, resulting in better alignment between the proximal response function and PBRF.

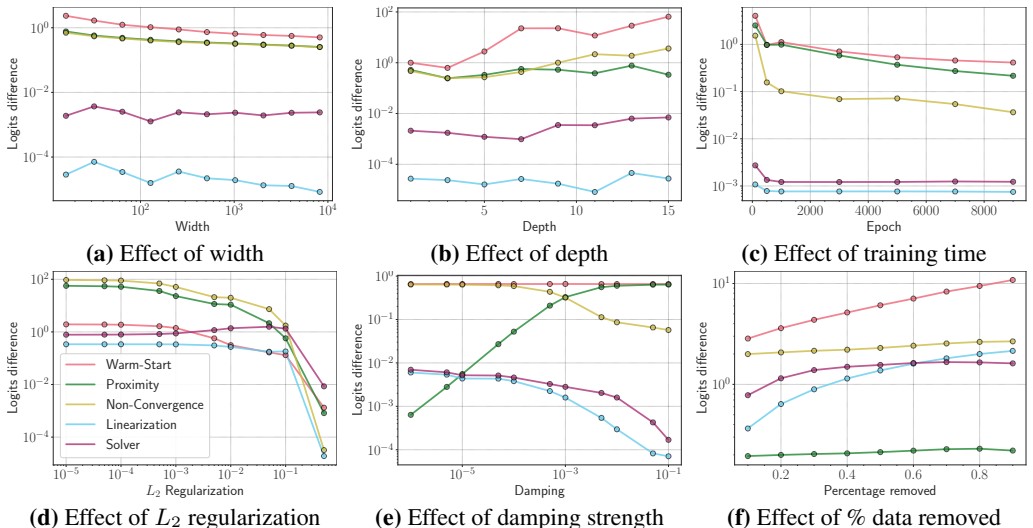

**Figure 5:** Ablations on how various factors affect the contribution of the gaps and errors to the discrepancy between influence approximation and LOO retraining.

**Weight Decay.** The weight decay allows the training objective to be better conditioned. Consequently, as weight decay increases, the training objective may act more as a strictly convex objective, resulting in a decrease in overall discrepancy for all components. Basu et al. [2020a] also found that the alignment between influence functions and LOO retraining increases as weight decay increases.

**Damping.** A higher damping term makes linear systems better conditioned, allowing solvers to find accurate solutions in fewer iterations [Demmel, 1997], thereby reducing the solver error. Furthermore, the higher proximity term keeps the parameters close to $\theta^s$, reducing the linearization error. On the other hand, increasing the effective proximity penalty directly increases the proximity gap.

**Percentage of Training Examples Removed.** As we remove more training examples from the dataset the PBRF becomes more non-linear and we observe a sharp increase in the linearization error. The cost landscape is also more likely to change as we remove more training examples, and we observe a corresponding increase in the warm-start gap.

# 7 Conclusion

In this paper, we investigate the sources of the discrepancy between influence functions and LOO retraining in neural networks. We decompose this difference into five distinct components: the warm-start gap, proximity gap, non-convergence gap, linearization error, and solver error. We empirically evaluate the contributions of each of these components on a wide variety of architectures and datasets and investigate how they change with factors such as network size and regularization. Our results show that the first three components are most responsible for the discrepancy between influence functions and LOO retraining. We further introduce the proximal Bregman response function (PBRF) to better capture the behavior of influence functions in neural networks. Compared to LOO retraining, the PBRF is more easily calculated and correlates better with influence functions, meaning it is an attractive alternative gold standard for evaluating influence functions. Although the PBRF may not necessarily align with LOO retraining, it can still be applied in many of the motivating use cases for influence functions. We conclude that influence functions in neural networks are not necessarily "fragile", but instead are giving accurate answers to a different question than is normally assumed.

# Acknowledgements

We would like to thank Pang Wei Koh for the helpful discussions. Resources used in this research were provided, in part, by the Province of Ontario, the Government of Canada through CIFAR, and companies sponsoring the Vector Institute (`www.vectorinstitute.ai/partners`).

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
