## Appendix

This appendix is structured as follows:

- In Section A, we provide an overview of the notation we use throughout the paper.
- In Section B, we provide derivations for influence functions, proximal response function, proximal Bregman response function, and linearized proximal response function.
- In Section C, we provide experimental details.
- In Section D, we provide additional experiment results.
- In Section E, we present the numerical results shown in Figure 4.
- In Section G, we provide an overview of CG and LiSSA algorithms.

## A    Table of Notation

Table 3 summarizes notations used in this paper.

| Notation | Description |
|---|---|
| $\mathcal{D}_{\text{train}}$ | Finite training dataset |
| $N$ | Number of training examples, $N = \|\mathcal{D}_{\text{train}}\|$ |
| $\mathbf{x}$ | Input of a data point |
| $\mathbf{t}$ | Target a data point |
| $\mathbf{z} = (\mathbf{x}, \mathbf{t})$ | Data point composed of an input $\mathbf{x}$ and a target $\mathbf{t}$ |
| $\mathbf{y} = f(\boldsymbol{\theta}, \mathbf{x})$ | Prediction of the network $f$ parameterized by the parameters $\boldsymbol{\theta}$ |
| $\boldsymbol{\theta}$ | Model parameters |
| $\boldsymbol{\theta}^{\star}$ | Optimal model parameters on the full training dataset $\mathcal{D}_{\text{train}}$ |
| $\boldsymbol{\theta}^{\star}_{-\mathbf{z}}$ | Optimal model parameters on the dataset with a data point $\mathbf{z}$ removed $\mathcal{D}_{\text{train}} - \{\mathbf{z}\}$ |
| $\mathcal{L}(\mathbf{y}, \mathbf{t})$ | Loss function (e.g., squared error or cross-entropy) |
| $\mathcal{J}(\boldsymbol{\theta})$ | Cost function on the full dataset $\mathcal{D}_{\text{train}}$ |
| $d$ | Number of model parameters |
| $\epsilon$ | Downweighting factor |
| $\mathcal{Q}_{-\mathbf{z}}(\boldsymbol{\theta}, \epsilon)$ | Downweighted cost defined as $\mathcal{Q}_{-\mathbf{z}}(\boldsymbol{\theta}, \epsilon) = \mathcal{J}(\boldsymbol{\theta}) - \mathcal{L}(f(\boldsymbol{\theta}, \mathbf{x}), \mathbf{t})\epsilon$ |
| $\boldsymbol{\theta}^{\star}_{-\mathbf{z},\epsilon}$ | Optimal model parameters of the downweighted cost $\mathcal{Q}_{-\mathbf{z}}(\boldsymbol{\theta}, \epsilon)$, where $\epsilon$ is fixed |
| $r^{\star}_{-\mathbf{z}}(\epsilon)$ | Response function |
| $\mathbf{J}_{\mathbf{y}\boldsymbol{\theta}^{\star}}$ | Parameter-output Jacobian at $\boldsymbol{\theta}^{\star}$ |
| $\mathbf{H}_{\mathbf{y}}$ | Hessian of the cost with respect to the network outputs |
| $\lambda$ | Damping strength |
| $r^{\star}_{-\mathbf{z},\text{damp}}(\epsilon)$ | Proximal response function |
| $\boldsymbol{\theta}^{s}$ | (Possibly non-converged) learned parameters |
| $\mathcal{D}_{\mathcal{L}^{(i)}}(\mathbf{y}, \mathbf{y}^{s})$ | The Bregman divergence $\mathcal{L}(\mathbf{y}, \mathbf{t}^{(i)}) - \mathcal{L}(\mathbf{y}^{s}, \mathbf{t}^{(i)}) - \nabla_{\mathbf{y}}\mathcal{L}(\mathbf{y}^{s}, \mathbf{t}^{(i)})^{\top}(\mathbf{y} - \mathbf{y}^{s})$ |
| $r^{b}_{-\mathbf{z},\text{damp}}(\epsilon)$ | Proximal Bregman response function |
| $r^{b}_{-\mathbf{z},\text{damp,lin}}(\epsilon)$ | Linearized proximal Bregman response function |
| $f_{\text{lin}}(\boldsymbol{\theta}, \mathbf{x})$ | Linearized network outputs with respect to the parameters $f_{\text{lin}}(\boldsymbol{\theta}, \mathbf{x}) = f(\boldsymbol{\theta}^{s}, \mathbf{x}) + \mathbf{J}_{\mathbf{y}\boldsymbol{\theta}^{s}}(\boldsymbol{\theta} - \boldsymbol{\theta}^{s})$ |
| $\mathbf{y}^{s} = f(\boldsymbol{\theta}^{s}, \mathbf{x})$ | Prediction of the network $f$ parameterized by the parameters $\boldsymbol{\theta}^{s}$ |
| $\mathcal{L}_{\text{quad}}(\mathbf{y}, \mathbf{t})$ | Second-order Taylor expansion of the loss around $\mathbf{y}^{s}$ |

**Table 3:** A summary of the notation used in this paper.

# B Derivations

## B.1 Influence Function Derivation

We provide a derivation of influence functions using the response function. We refer readers to Van der Vaart [2000] and Koh and Liang [2017] for a more general derivation of influence functions.

Let $\mathbf{z} = (\mathbf{x}, \mathbf{t}) \in \mathcal{D}_{\text{train}}$ be a training example we are interested in downweighting. Recall that the downweighted objective is defined as:

$$\mathcal{Q}_{-\mathbf{z}}(\boldsymbol{\theta}, \epsilon) = \mathcal{J}(\boldsymbol{\theta}) - \mathcal{L}(f(\boldsymbol{\theta}, \mathbf{x}), \mathbf{t})\epsilon. \tag{11}$$

We further let $\epsilon_0 = 0$ and $\boldsymbol{\theta}^\star$ be the optimal parameters such that $\nabla_{\boldsymbol{\theta}} \mathcal{Q}_{-\mathbf{z}}(\boldsymbol{\theta}^\star, \epsilon_0) = \mathbf{0}$. Here, the optimal parameters $\boldsymbol{\theta}^\star$ is the solution that minimizes the cost function $\mathcal{J}(\cdot)$. We assume that the downweighted objective is twice continuously differentiable and strongly convex in the parameters $\boldsymbol{\theta}$ at $\epsilon_0$. Note that if we assume the strong convexity of the loss function, the downweighted objective is only guaranteed to be strongly convex when $\epsilon \leq 1/N$.

By the Implicit Function Theorem, these exists a unique continuously differentiable response function $r^\star_{-\mathbf{z}} : \mathcal{U}_0 \to \mathbb{R}^d$ defined on a neighborhood $\mathcal{U}_0$ of $\epsilon_0$ such that $r^\star_{-\mathbf{z}}(\epsilon_0) = \boldsymbol{\theta}^\star$ and:

$$\nabla_{\boldsymbol{\theta}} \mathcal{Q}_{-\mathbf{z}}(r^\star_{-\mathbf{z}}(\epsilon), \epsilon) = \mathbf{0} \tag{12}$$

for all $\epsilon \in \mathcal{U}_0$. By taking the derivative with respect to the downweighting factor $\epsilon$, we get:

$$\mathbf{0} = \frac{\mathrm{d}}{\mathrm{d}\epsilon} \left( \nabla_{\boldsymbol{\theta}} \mathcal{Q}_{-\mathbf{z}}(r^\star_{-\mathbf{z}}(\epsilon), \epsilon) \right) = \nabla^2_{\boldsymbol{\theta}} \mathcal{Q}_{-\mathbf{z}}(r^\star_{-\mathbf{z}}(\epsilon), \epsilon) \frac{\mathrm{d}r^\star_{-\mathbf{z}}}{\mathrm{d}\epsilon}(\epsilon) + \nabla^2_{\boldsymbol{\theta}, \epsilon} \mathcal{Q}_{-\mathbf{z}}(r^\star_{-\mathbf{z}}(\epsilon), \epsilon) \tag{13}$$

for all $\epsilon \in \mathcal{U}_0$. The Jacobian of the response function at $\epsilon_0$ can further be expressed as:

$$\mathbf{0} = \left( \nabla^2_{\boldsymbol{\theta}} \mathcal{J}(\boldsymbol{\theta}^\star) \right) \left( \frac{\mathrm{d}r^\star_{-\mathbf{z}}}{\mathrm{d}\epsilon} \bigg|_{\epsilon = \epsilon_0} \right) - \nabla_{\boldsymbol{\theta}} \mathcal{L}(f(\boldsymbol{\theta}^\star, \mathbf{x}), \mathbf{t}), \tag{14}$$

where we used these two equalities:

$$\nabla^2_{\boldsymbol{\theta}} \mathcal{Q}_{-\mathbf{z}}(r^\star_{-\mathbf{z}}(\epsilon_0), \epsilon_0) = \nabla^2_{\boldsymbol{\theta}} \mathcal{J}(\boldsymbol{\theta}^\star) \tag{15}$$

$$\nabla^2_{\boldsymbol{\theta}, \epsilon} \mathcal{Q}_{-\mathbf{z}}(r^\star_{-\mathbf{z}}(\epsilon_0), \epsilon_0) = -\nabla_{\boldsymbol{\theta}} \mathcal{L}(f(\boldsymbol{\theta}^\star, \mathbf{x}), \mathbf{t}) \tag{16}$$

Rearranging Eqn. 14, we have:

$$\frac{\mathrm{d}r^\star_{-\mathbf{z}}}{\mathrm{d}\epsilon} \bigg|_{\epsilon = \epsilon_0} = \left( \nabla^2_{\boldsymbol{\theta}} \mathcal{J}(\boldsymbol{\theta}^\star) \right)^{-1} \nabla_{\boldsymbol{\theta}} \mathcal{L}(f(\boldsymbol{\theta}^\star, \mathbf{x}), \mathbf{t}), \tag{17}$$

where the Hessian $\nabla^2_{\boldsymbol{\theta}} \mathcal{J}(\boldsymbol{\theta}^\star)$ is invertible by strong convexity of our downweighted objective at $\epsilon_0$. Influence functions approximate the response function with a first-order Taylor expansion at $\epsilon_0$:

$$r^\star_{-\mathbf{z}, \text{lin}}(\epsilon) = r^\star_{-\mathbf{z}}(\epsilon_0) + \frac{\mathrm{d}r^\star_{-\mathbf{z}}}{\mathrm{d}\epsilon} \bigg|_{\epsilon = \epsilon_0} (\epsilon - \epsilon_0) = \boldsymbol{\theta}^\star + (\nabla^2_{\boldsymbol{\theta}} \mathcal{J}(\boldsymbol{\theta}^\star))^{-1} \nabla_{\boldsymbol{\theta}} \mathcal{L}(f(\boldsymbol{\theta}^\star, \mathbf{x}), \mathbf{t})\epsilon. \tag{18}$$

To approximate the optimal parameters trained without a data point $\mathbf{z}$, we can substitute $\epsilon = 1/N$ as follows:

$$r^\star_{-\mathbf{z}, \text{lin}}(1/N) = \boldsymbol{\theta}^\star + \frac{1}{N} (\nabla^2_{\boldsymbol{\theta}} \mathcal{J}(\boldsymbol{\theta}^\star))^{-1} \nabla_{\boldsymbol{\theta}} \mathcal{L}(f(\boldsymbol{\theta}^\star, \mathbf{x}), \mathbf{t}). \tag{19}$$

Influence functions can further approximate the loss at a particular test point $\mathbf{z}_{\text{test}} = (\mathbf{x}_{\text{test}}, \mathbf{t}_{\text{test}})$ (or test loss) when a training example $\mathbf{z}$ is eliminated from the training set using the chain rule [Koh and Liang, 2017]:

$$\mathcal{L}(f(r^\star_{-\mathbf{z}, \text{lin}}(1/N), \mathbf{x}_{\text{test}}), \mathbf{t}_{\text{test}})$$
$$\approx \mathcal{L}(f(\boldsymbol{\theta}^\star, \mathbf{x}_{\text{test}}), \mathbf{t}_{\text{test}}) + \frac{1}{N} \nabla_{\boldsymbol{\theta}} \mathcal{L}(f(\boldsymbol{\theta}^\star, \mathbf{x}_{\text{test}}), \mathbf{t}_{\text{test}})^\top \frac{\mathrm{d}r^\star_{-\mathbf{z}}}{\mathrm{d}\epsilon} \bigg|_{\epsilon = \epsilon_0} \tag{20}$$
$$\approx \mathcal{L}(f(\boldsymbol{\theta}^\star, \mathbf{x}_{\text{test}}), \mathbf{t}_{\text{test}}) + \frac{1}{N} \nabla_{\boldsymbol{\theta}} \mathcal{L}(f(\boldsymbol{\theta}^\star, \mathbf{x}_{\text{test}}), \mathbf{t}_{\text{test}})^\top (\nabla^2_{\boldsymbol{\theta}} \mathcal{J}(\boldsymbol{\theta}^\star))^{-1} \nabla_{\boldsymbol{\theta}} \mathcal{L}(f(\boldsymbol{\theta}^\star, \mathbf{x}), \mathbf{t}).$$

## B.2 Proximal Response Function Derivation

Let $\mathbf{z} = (\mathbf{x}, \mathbf{t}) \in \mathcal{D}_{\text{train}}$ be a training example we are interested in downweighting. Recall that the proximal response function (in Eqn. 7) is defined as:

$$r^\star_{-\mathbf{z},\text{damp}}(\epsilon) = \underset{\boldsymbol{\theta} \in \mathbb{R}^d}{\arg\min} \, \mathcal{Q}_{-\mathbf{z}}(\boldsymbol{\theta}, \epsilon) + \frac{\lambda}{2}\|\boldsymbol{\theta} - \boldsymbol{\theta}^\star\|^2, \tag{21}$$

where $\boldsymbol{\theta}^\star$ is the optimal parameters that minimize the cost function and $\lambda > 0$ is a damping term. Here, we show that influence estimations with a damping term correspond to a first-order Taylor approximation of the proximal response function. Let $\lambda > 0$ be some damping term and $\boldsymbol{\theta}^\star$ be the solution that minimizes the cost function. Let $\epsilon_0 = 0$ and assume that the downweighted objective is convex in the parameters $\boldsymbol{\theta}$ at $\epsilon_0$. By the Implicit Function Theorem, we can guarantee the existence of the proximal response function $r^\star_{-\mathbf{z},\text{damp}} : \mathcal{U}_0 \to \mathbb{R}^d$ defined on a neighborhood $\mathcal{U}_0$ of $\epsilon_0$ which satisfies $r^\star_{-\mathbf{z},\text{damp}}(\epsilon_0) = \boldsymbol{\theta}^\star$ and:

$$\nabla_{\boldsymbol{\theta}} \mathcal{Q}_{-\mathbf{z}}(r^\star_{-\mathbf{z},\text{damp}}(\epsilon), \epsilon) + \lambda\left(r_{-\mathbf{z},\text{damp}}(\epsilon)^\star - \boldsymbol{\theta}^\star\right) = \mathbf{0} \tag{22}$$

for all $\epsilon$ in some neighborhood $\mathcal{U}_0$ of $\epsilon_0$. Then, differentiating Eqn. 22 with respect to the downweighting factor $\epsilon$ equates:

$$\mathbf{0} = \frac{\mathrm{d}}{\mathrm{d}\epsilon} \left(\nabla_{\boldsymbol{\theta}} \mathcal{Q}_{-\mathbf{z}}(r^\star_{-\mathbf{z},\text{damp}}(\epsilon), \epsilon) + \lambda\left(r^\star_{-\mathbf{z},\text{damp}}(\epsilon) - \boldsymbol{\theta}^\star\right)\right) \tag{23}$$

$$= \nabla^2_{\boldsymbol{\theta}} \mathcal{Q}_{-\mathbf{z}}(r^\star_{-\mathbf{z},\text{damp}}(\epsilon), \epsilon)\frac{\mathrm{d}r^\star_{-\mathbf{z},\text{damp}}}{\mathrm{d}\epsilon}(\epsilon) + \nabla^2_{\boldsymbol{\theta},\epsilon} \mathcal{Q}_{-\mathbf{z}}(r^\star_{-\mathbf{z},\text{damp}}(\epsilon), \epsilon) + \lambda\frac{\mathrm{d}r^\star_{-\mathbf{z},\text{damp}}}{\mathrm{d}\epsilon}(\epsilon) \tag{24}$$

for all $\epsilon \in \mathcal{U}_0$. Evaluating the response Jacobian at $\epsilon_0$ and rearranging the terms in Eqn. 24:

$$\frac{\mathrm{d}r^\star_{-\mathbf{z},\text{damp}}}{\mathrm{d}\epsilon}\bigg|_{\epsilon=\epsilon_0} = \left(\nabla^2_{\boldsymbol{\theta}} \mathcal{J}(\boldsymbol{\theta}^\star) + \lambda\mathbf{I}\right)^{-1} \nabla_{\boldsymbol{\theta}} \mathcal{L}(f(\boldsymbol{\theta}^\star, \mathbf{x}), \mathbf{t}). \tag{25}$$

Hence, a first-order Taylor approximation of the proximal response function is equivalent to influence functions with a damping term $\lambda\mathbf{I}$ added:

$$r^\star_{-\mathbf{z},\text{damp,lin}}(\epsilon) = r^\star_{-\mathbf{z},\text{damp}}(\epsilon_0) + \frac{\mathrm{d}r^\star_{-\mathbf{z},\text{damp}}}{\mathrm{d}\epsilon}\bigg|_{\epsilon=\epsilon_0}(\epsilon - \epsilon_0) \tag{26}$$

$$= \boldsymbol{\theta}^\star + \left(\nabla^2_{\boldsymbol{\theta}} \mathcal{J}(\boldsymbol{\theta}^\star) + \lambda\mathbf{I}\right)^{-1}\nabla_{\boldsymbol{\theta}} \mathcal{L}(f(\boldsymbol{\theta}^\star, \mathbf{x}), \mathbf{t})\epsilon. \tag{27}$$

When a damping term is used in influence functions, they approximate LOO retraining scheme with the proximity term added to the downweighted objective.

## B.3 Proximal Bregman Response Function Derivation

As opposed to the derivation from Appendix B.1, we consider computing influence functions on parameters $\boldsymbol{\theta}^s$ that have not necessarily converged. When the parameters have not fully converged, a warm-start retraining with the downweighted objective defined in Eqn. 11 would simply minimize the cost function in the first term and reflect the effect of training longer, rather than the effect of removing a training example.

Let $\mathbf{z} = (\mathbf{x}, \mathbf{t}) \in \mathcal{D}_{\text{train}}$ be a training example we are interested in downweighting. We assume that the loss function is convex as a function of the network outputs which hold for commonly used loss functions. We replace the cost function in the downweighted objective with a term that penalizes mismatch to the predictions made by the current parameters $\boldsymbol{\theta}^s$ and define the proximal Bregman downweighted objective as:

$$\mathcal{Q}^b_{-\mathbf{z}}(\boldsymbol{\theta}, \epsilon) = \frac{1}{N}\sum_{i=1}^{N} D_{\mathcal{L}^{(i)}}(f(\boldsymbol{\theta}, \mathbf{x}^{(i)}), f(\boldsymbol{\theta}^s, \mathbf{x}^{(i)})) - \mathcal{L}(f(\boldsymbol{\theta}, \mathbf{x}), \mathbf{t})\epsilon + \frac{\lambda}{2}\|\boldsymbol{\theta} - \boldsymbol{\theta}^s\|, \tag{28}$$

where $D_{\mathcal{L}^{(i)}}(\cdot, \cdot)$ is the Bregman divergence defined as:

$$D_{\mathcal{L}^{(i)}}(\mathbf{y}, \mathbf{y}^s) = \mathcal{L}(\mathbf{y}, \mathbf{t}^{(i)}) - \mathcal{L}(\mathbf{y}^s, \mathbf{t}^{(i)}) - \nabla_{\mathbf{y}}\mathcal{L}(\mathbf{y}^s, \mathbf{t}^{(i)})^\top(\mathbf{y} - \mathbf{y}^s). \tag{29}$$

Because of our convexity assumption, the Bregman divergence term in Eqn. 28 is non-negative and is 0 when $\boldsymbol{\theta} = \boldsymbol{\theta}^s$. Hence, the parameters $\boldsymbol{\theta}^s$ is optimal at $\epsilon = 0$ on the proximal Bregman downweighted objective although it is not optimal on the cost function. Accordingly, we define the proximal Bregman response function (PBRF) as follows:

$$r^b_{-\mathbf{z},\text{damp}}(\epsilon) = \underset{\boldsymbol{\theta} \in \mathbb{R}^d}{\arg\min} \frac{1}{N} \sum_{i=1}^{N} D_{\mathcal{L}^{(i)}}(f(\boldsymbol{\theta}, \mathbf{x}^{(i)}), f(\boldsymbol{\theta}^s, \mathbf{x}^{(i)})) - \mathcal{L}(f(\boldsymbol{\theta}, \mathbf{x}), \mathbf{t})\epsilon + \frac{\lambda}{2}\|\boldsymbol{\theta} - \boldsymbol{\theta}^s\|, \tag{30}$$

where we assume that the proximal Bregman downweighted objective is strongly convex at $\epsilon_0$ and the solution to the Bregman downweighted objective is unique. Letting $\epsilon_0 = 0$, the Bregman response function satisfies $r^b_{-\mathbf{z},\text{damp}}(\epsilon_0) = \boldsymbol{\theta}^s$ and:

$$\nabla_{\boldsymbol{\theta}} \mathcal{Q}^b_{-\mathbf{z}}(r^b_{-\mathbf{z},\text{damp}}(\epsilon), \epsilon) = \mathbf{0}. \tag{31}$$

for all downweighting factor $\epsilon$ in some neighborhood of $\epsilon_0$. By taking the derivative with respect to the downweighting factor $\epsilon$, we get:

$$\mathbf{0} = \frac{\mathrm{d}}{\mathrm{d}\epsilon} \left( \nabla_{\boldsymbol{\theta}} \mathcal{Q}^b_{-\mathbf{z}}(r^b_{-\mathbf{z},\text{damp}}(\epsilon), \epsilon) \right) \tag{32}$$

$$= \nabla^2_{\boldsymbol{\theta}} \mathcal{Q}^b_{-\mathbf{z}}(r^b_{-\mathbf{z},\text{damp}}(\epsilon), \epsilon) \frac{\mathrm{d}r^b_{-\mathbf{z},\text{damp}}}{\mathrm{d}\epsilon}(\epsilon) + \nabla^2_{\boldsymbol{\theta},\epsilon} \mathcal{Q}^b_{-\mathbf{z}}(r^b_{-\mathbf{z},\text{damp}}(\epsilon), \epsilon). \tag{33}$$

For linear models, where the parameter-output Jacobian is constant, the Jacobian of the response function at $\epsilon_0$ can further be expressed as:

$$\mathbf{0} = \left( \nabla^2_{\boldsymbol{\theta}} \mathcal{J}(\boldsymbol{\theta}^s) + \lambda \mathbf{I} \right) \left( \frac{\mathrm{d}r^b_{-\mathbf{z},\text{damp}}}{\mathrm{d}\epsilon} \bigg|_{\epsilon=\epsilon_0} \right) - \nabla_{\boldsymbol{\theta}} \mathcal{L}(f(\boldsymbol{\theta}^s, \mathbf{x}), \mathbf{t}), \tag{34}$$

where we used these two equalities:

$$\nabla^2_{\boldsymbol{\theta}} \mathcal{Q}^b_{-\mathbf{z}}(r^b_{-\mathbf{z},\text{damp}}(\epsilon_0), \epsilon_0) = \nabla^2_{\boldsymbol{\theta}} \mathcal{J}(\boldsymbol{\theta}^s) + \lambda \mathbf{I} \tag{35}$$

$$\nabla^2_{\boldsymbol{\theta},\epsilon} \mathcal{Q}^b_{-\mathbf{z}}(r^b_{-\mathbf{z},\text{damp}}(\epsilon_0), \epsilon_0) = -\nabla_{\boldsymbol{\theta}} \mathcal{L}(f(\boldsymbol{\theta}^s, \mathbf{x}), \mathbf{t}) \tag{36}$$

Rearranging the terms in Eqn. 34, the Jacobian of the PBRF at $\epsilon_0$ can be expressed as:

$$\frac{\mathrm{d}r^b_{-\mathbf{z}}}{\mathrm{d}\epsilon} \bigg|_{\epsilon=\epsilon_0} = \left( \nabla^2_{\boldsymbol{\theta}} \mathcal{J}(\boldsymbol{\theta}^s) + \lambda \mathbf{I} \right)^{-1} \nabla_{\boldsymbol{\theta}} \mathcal{L}(f(\boldsymbol{\theta}^s, \mathbf{x}), \mathbf{t}). \tag{37}$$

Note that both Hessian and gradient are computed on the final parameters $\boldsymbol{\theta}^s$ instead of the optimal parameters $\boldsymbol{\theta}^\star$. Hence, influence functions at the non-converged parameters $\boldsymbol{\theta}^s$ (with a damping) can be seen as an approximation to the PBRF rather than an approximation to LOO retraining.

## B.4 Linearized Proximal Bregman Response Function Derivation

We show that the linearized proximal Bregman response function (PBRF) is equivalent to the influence estimation with the Gauss-Newton Hessian approximation and a damping term $\lambda > 0$. Let $\boldsymbol{\theta}^s \in \mathbb{R}^d$ be a possibly non-converged learned parameters, $\mathbf{z} = (\mathbf{x}, \mathbf{t}) \in \mathcal{D}_{\text{train}}$ be a training example we want to downweight, and $\lambda > 0$ be a damping term. Recall that the linearized PBRF is defined as:

$$r^b_{-\mathbf{z},\text{damp,lin}}(\epsilon) = \underset{\boldsymbol{\theta} \in \mathbb{R}^d}{\arg\min} \frac{1}{N} \sum_{i=1}^{N} D_{\mathcal{L}^{(i)}_{\text{quad}}}(f_{\text{lin}}(\boldsymbol{\theta}, \mathbf{x}^{(i)}), f(\boldsymbol{\theta}^s, \mathbf{x}^{(i)}))$$
$$- \nabla_{\boldsymbol{\theta}} \mathcal{L}(f(\boldsymbol{\theta}^s, \mathbf{x}), \mathbf{t})^\top \boldsymbol{\theta} \epsilon + \frac{\lambda}{2}\|\boldsymbol{\theta} - \boldsymbol{\theta}^s\|^2, \tag{38}$$

where $\mathcal{L}_{\text{quad}}(\cdot, \cdot)$ and $f_{\text{lin}}(\boldsymbol{\theta}, \mathbf{x})$ are defined as:

$$\mathcal{L}_{\text{quad}}(\mathbf{y}, \mathbf{t}) = \mathcal{L}(\mathbf{y}^s, \mathbf{t}) + \nabla_{\mathbf{y}} \mathcal{L}(\mathbf{y}^s, \mathbf{t})^\top (\mathbf{y} - \mathbf{y}^s) + (\mathbf{y} - \mathbf{y}^s)^\top \nabla^2_{\mathbf{y}} \mathcal{L}(\mathbf{y}^s, \mathbf{t})(\mathbf{y} - \mathbf{y}^s). \tag{39}$$

$$f_{\text{lin}}(\boldsymbol{\theta}, \mathbf{x}^{(i)}) = f(\boldsymbol{\theta}^s, \mathbf{x}^{(i)}) + \mathbf{J}_{\mathbf{y}^{(i)}\boldsymbol{\theta}^s}(\boldsymbol{\theta} - \boldsymbol{\theta}^s). \tag{40}$$

Here, $\mathbf{y}^s$ is the prediction of the network parameterized by $\boldsymbol{\theta}^s$ and $\mathbf{J}_{\mathbf{y}^{(i)}\boldsymbol{\theta}^s}$ is the parameter-output Jacobian of the $i$-th training example. We further assume that the loss function is convex as a function of the network outputs which hold for commonly used loss functions. The first Bregman divergence term in Eqn. 38 can be expressed as:

$$D_{\mathcal{L}_{\text{quad}}^{(i)}}(f_{\text{lin}}(\boldsymbol{\theta}, \mathbf{x}^{(i)}), \mathbf{y}^s) = \nabla_{\mathbf{y}}\mathcal{L}(\mathbf{y}^s, \mathbf{t}^{(i)})^\top \mathbf{J}_{\mathbf{y}^{(i)}\boldsymbol{\theta}^s}(\boldsymbol{\theta} - \boldsymbol{\theta}^s) \tag{41}$$

$$+ (\boldsymbol{\theta} - \boldsymbol{\theta}^s)^\top \mathbf{J}_{\mathbf{y}^{(i)}\boldsymbol{\theta}^s}^\top \nabla_{\mathbf{y}}^2 \mathcal{L}(\mathbf{y}^s, \mathbf{t}^{(i)}) \mathbf{J}_{\mathbf{y}^{(i)}\boldsymbol{\theta}^s}(\boldsymbol{\theta} - \boldsymbol{\theta}^s) \tag{42}$$

$$- \nabla_{\mathbf{y}}\mathcal{L}(\mathbf{y}^s, \mathbf{t}^{(i)})^\top \mathbf{J}_{\mathbf{y}^{(i)}\boldsymbol{\theta}^s}(\boldsymbol{\theta} - \boldsymbol{\theta}^s) \tag{43}$$

$$= (\boldsymbol{\theta} - \boldsymbol{\theta}^s)^\top \mathbf{J}_{\mathbf{y}^{(i)}\boldsymbol{\theta}^s}^\top \nabla_{\mathbf{y}}^2 \mathcal{L}(\mathbf{y}^s, \mathbf{t}^{(i)}) \mathbf{J}_{\mathbf{y}^{(i)}\boldsymbol{\theta}^s}(\boldsymbol{\theta} - \boldsymbol{\theta}^s). \tag{44}$$

Now, taking the gradient of linearized PBRF objective with respect to the parameters $\boldsymbol{\theta}$ and setting it equal to $\mathbf{0}$, we get:

$$\mathbf{0} = \frac{1}{N}\sum_{i=1}^N \left( \mathbf{J}_{\mathbf{y}^{(i)}\boldsymbol{\theta}^s}^\top \nabla_{\mathbf{y}}^2 \mathcal{L}(\mathbf{y}^s, \mathbf{t}^{(i)}) \mathbf{J}_{\mathbf{y}^{(i)}\boldsymbol{\theta}^s}(\boldsymbol{\theta} - \boldsymbol{\theta}^s) \right) - \nabla_{\boldsymbol{\theta}}\mathcal{L}(f(\boldsymbol{\theta}^s, \mathbf{x}), \mathbf{t})\epsilon + \lambda(\boldsymbol{\theta} - \boldsymbol{\theta}^s)$$
$$= \mathbf{J}_{\mathbf{y}\boldsymbol{\theta}^s}^\top \mathbf{H}_{\mathbf{y}}^s \mathbf{J}_{\mathbf{y}\boldsymbol{\theta}^s}(\boldsymbol{\theta} - \boldsymbol{\theta}^s) - \nabla_{\boldsymbol{\theta}}\mathcal{L}(f(\boldsymbol{\theta}^s, \mathbf{x}), \mathbf{t})\epsilon + \lambda(\boldsymbol{\theta} - \boldsymbol{\theta}^s), \tag{45}$$

where $\mathbf{H}_{\mathbf{y}}^s$ is the Hessian of the loss with respect to the network outputs evaluated at $\mathbf{y}^s$. Rearranging the terms, we get:

$$\boldsymbol{\theta}_{\text{lin, PBRF}}^\star = \boldsymbol{\theta}^s + \left( \mathbf{J}_{\mathbf{y}\boldsymbol{\theta}^s}^\top \mathbf{H}_{\mathbf{y}}^s \mathbf{J}_{\mathbf{y}\boldsymbol{\theta}^s} + \lambda \mathbf{I} \right)^{-1} \nabla_{\boldsymbol{\theta}}\mathcal{L}(f(\boldsymbol{\theta}^s, \mathbf{x}), \mathbf{t})\epsilon, \tag{46}$$

where $\boldsymbol{\theta}_{\text{lin, PBRF}}^\star$ is the optimal solution to the linearized PBRF objective. Note that the Gauss-Newton Hessian (GNH) $\mathbf{G}^s = \mathbf{J}_{\mathbf{y}\boldsymbol{\theta}^s}^\top \mathbf{H}_{\mathbf{y}}^s \mathbf{J}_{\mathbf{y}\boldsymbol{\theta}^s}$ is positive semidefinite (assuming that the loss function is convex as a function of network outputs) and an addition of damping term guarantees the invertibility. Therefore, the optimal solution to Eqn. 46 is equivalent to the influence estimation with the GNH approximation and a damping term $\lambda$.

## C   Experimental Details

### C.1   Computing Environment

All experiments were implemented using the PyTorch [Paszke et al., 2019] and JAX [Bradbury et al., 2018] frameworks and we ran all experiments on NVIDIA P100 GPUs.

### C.2   Experiment Set-up

In all experiments, we first trained the base network (initialized with some $\boldsymbol{\theta}^0$) with the entire dataset for $K$ epochs to obtain the base parameters $\boldsymbol{\theta}^s$. We then select 20 random data points $\mathbf{z}_t \in \mathcal{D}_{\text{train}}$ from the training dataset and computed the additional parameters for each $t$ as follows:

1. **Cold optimum.** We retrained the network for $K + K/2$ epochs with the initialization $\boldsymbol{\theta}^0$. To minimize the effect of stochasticity in stochastic gradient-based optimizer, we further used the same batch order used for training the base network for the first $K$ epochs.

2. **Warm optimum.** We retrained the network for $K/2$ epochs with the initialization $\boldsymbol{\theta}^s$.

3. **Proximal warm optimum.** We retrained the network for $K/2$ epochs with the initialization $\boldsymbol{\theta}^s$ using the objective defined in Eqn. 7.

4. **Proximal Bregman warm optimum.** We retrained the network for $K/2$ epochs with the initialization $\boldsymbol{\theta}^s$ using the objective defined in Eqn. 8.

5. **Linearized proximal Bregman warm optimum.** We retrained the network for $K/2$ epochs with the initialization $\boldsymbol{\theta}^s$ using the objective defined in Eqn. 10.

6. **Influence estimation.** We used the `LiSSA` algorithm on the base parameters $\boldsymbol{\theta}^s$ with the Gauss-Newton Hessian approximation (Eqn. 6).

The "warm-start gap" refers to the discrepancy between cold-start and warm-start optima. The "proximity gap" refers to the discrepancy between warm-start and proximal warm-start optima. The "non-convergence gap" denotes the discrepancy between proximal warm-start and proximal Bregman warm-start optima. The "linearization error" represents the discrepancy between proximal Bregman warm-start and linearized proximal Bregman warm-start optima. Lastly, the "solver error" denotes the discrepancy between linearized proximal Bregman warm optimum and influence estimation with the `LiSSA` algorithm.

We treated the scaling in the `LiSSA` algorithm as a separate hyperparameter [Koh and Liang, 2017] and tuned the scaling in the range {10, 25, 50, 100, 150, 200, 250, 300, 400, 500} so that the algorithm converges.

### C.3 Influence Misalignment Decomposition

**Logistic Regression.** We used Cancer and Diabetes classification datasets from the UCI collection [Dua and Graff, 2017]. In training, we normalized the input features to have zero mean and unit variance. We trained the model using `L-BFGS` with $L_2$ regularization of 0.01 and damping term of $\lambda = 0.001$.

**Multilayer Perceptron.** For regression experiments, we used 2-hidden layer MLP with 128 hidden units. For both Concrete and Energy datasets, we normalized the input features and targets to have a zero mean and unit variance. For classification experiments with 10% of MNIST [Deng, 2012] and FashionMNIST [Xiao et al., 2017] datasets, we used 2-hidden layer MLP with a hidden unit dimension of 1024. For both regression and classification experiments, we used a batch size of 128 and trained the base network for 1000 epochs using SGD. While we did not use any $L_2$ regularization, we set the damping strength to $\lambda = 0.001$.

We conducted the hyperparameter searches over the learning rates for the base model, making choices based on the final validation loss. We swept over the learning rates {1.0, 0.3, 0.1, 0.03, 0.01, 0.003, 0.001}. We used a learning rate decayed by a factor of 10 for computing PBRF and linearized PBRF. We set the recursion depth to 5000 and the number of repeat to 5 for the `LiSSA` algorithm.

**Autoencoder.** We used the same experimental set-up from Martens and Grosse [2015]. The loss function was the binary cross-entropy and the $L_2$ regularization with strength $5 \cdot 10^{-5}$ was added to the cost function. The layer widths for the autoencoder were [784, 1000, 500, 250, 30, 250, 500, 1000, 784] and we used sigmoid activation functions. We trained the network for 1000 epochs on the full MNIST dataset with SGDm (SGD with momentum) and set the batch size to 1024. The damping term was set to $\lambda = 0.001$.

We conducted the hyperparameter searches for the base model making choices based on the final validation loss. We kept the momentum to 0.9 and swept over the learning rates 1, 0.3, 0.1, 0.03, 0.01, 0.003, 0.001. We used a learning rate decayed by a factor of 10 for computing PBRF and linearized PBRF. We set the recursion depth to 10000 and the number of repeat to 5 for the `LiSSA` algorithm.

**Convolution Neural Networks.** We trained LeNet [Lecun et al., 1998], AlexNet [Krizhevsky et al., 2012], VGG13 Simonyan and Zisserman [2014], and ResNet-20 [He et al., 2015] on 10% of MNIST dataset and the full CIFAR10 [Krizhevsky, 2009] dataset. For the MNIST experiment, we kept the learning rate fixed, while for CIFAR10 experiment, we decayed the initial learning rate by a factor of 5 at epochs 60, 120, 160. We used $L_2$ regularization of $5 \cdot 10^{-4}$ and a damping factor of $\lambda = 0.001$. For both datasets, we trained the base network for 200 epochs with the batch size of 128.

We used 10% of the MNIST dataset for the test loss correlation experiments in Table 2 and computed the approximated test loss with Eqn. 20 on randomly selected test examples. We conducted the hyperparameter searches for the base model making choices based on the final validation accuracy. We fixed the momentum to 0.9 and swept over the initial learning rates {1.0, 0.3, 0.1, 0.03, 0.01, 0.003, 0.001}. We used a learning rate decayed by a factor of 10 for computing PBRF and linearized PBRF. We set the recursion depth to 10000 and the number of repeat to 5 for the `LiSSA` algorithm.

**Transformer.** We trained a 2-layer Transformer language model on the Penn Treebank (PTB) dataset [Marcus et al., 1993]. The number of hidden dimensions was set to 256 and the number of attention heads was set to 2. We trained the model with Adam for 10 epochs. We set the batch size to 20 and a damping term of $\lambda = 0.01$. We conducted the hyperparameter searches for the base model making choices based on the final validation perplexity. We swept over the learning rates {0.03, 0.01,

| Model | Dataset | Warm-Start | Proximity | Non-Convergence | Linearization | Solver |
|-------|---------|-----------|-----------|-----------------|---------------|--------|
| MLP | Concrete | $0.079 \pm 0.008$ | $0.002 \pm 0.000$ | $\mathbf{0.082} \pm 0.001$ | $0.000 \pm 0.000$ | $0.017 \pm 0.026$ |
| | Energy | $\mathbf{0.016} \pm 0.001$ | $0.001 \pm 0.000$ | $\mathbf{0.016} \pm 0.001$ | $0.000 \pm 0.000$ | $0.000 \pm 0.000$ |
| | MNIST | $\mathbf{0.208} \pm 0.732$ | $0.006 \pm 0.024$ | $0.211 \pm 0.745$ | $0.000 \pm 0.000$ | $0.003 \pm 0.006$ |
| | FashionMNIST | $\mathbf{2.968} \pm 0.545$ | $0.385 \pm 0.028$ | $0.408 \pm 0.033$ | $0.000 \pm 0.001$ | $0.009 \pm 0.014$ |

**Table 4:** Decomposition of the discrepancy between influence functions (without the Gauss-Newton Hessian approximation) and LOO retraining into (1) warm-start gap, (2) proximal gap, (3) non-convergence gap, (4) linearization error, and (5) solver error for each model and dataset. Different from Table 5, we computed influence functions **without** the Gauss-Newton Hessian approximation. The size of each component is measured by the $L_2$ distance between the networks' outputs on the training dataset.

0.003, 0.001, 0.0003, 0.0001} for training the base model and set the recursion depth to 5000 and the number of repeat to 5 for the `LiSSA` algorithm.

### C.4    Factors in Influence Misalignment

The experiment in Section 6.2 was conducted using 10% of the MNIST dataset. We trained 2-hidden layer MLP composed of 1024 hidden units for 1000 epochs using SGD. We used the batch size of 128 and conducted the hyperparameter searches for each model making choices based on the final validation loss. We swept over the learning rates {1.0, 0.3, 0.1, 0.03, 0.01, 0.003, 0.001, 0.0003, 0.0001}.

To see how the gaps and errors change as we increase the width of the network, we repeated the experiments with all widths in a set {16, 32, 64, 128, 256, 512, 1024, 2048, 4096, 8192} while keeping the depth fixed to 2. To see the effect of increase in the depth of the network, we fixed the width to 1024 and computed gaps and errors with depth {1, 3, 5, 7, 9, 11, 13, 15}.

While keeping all configurations the same (2-hidden layer MLP with 1024 hidden units), we computed the decomposition by changing the total number of epochs to train the base model in the range {100, 500, 1000, 3000, 5000, 7000, 9000}, changing the strength of the weight decay {0.5, 0.1, 0.05, 0.01, 0.005, 0.001, 0.0005, 0.0001, 0.00005, 0.00001}, and changing the strength of the damping term in the range {0.5, 0.1, 0.05, 0.01, 0.005, 0.001, 0.0005, 0.0001, 0.00005, 0.00001, 0.000005, 0.000001}. Lastly, we modified the downweighted objective to remove a group of training examples (rather than a single training example) and altered the percentage we remove the training dataset in range {0.1, 0.2, 0.3, 0.4, 0.5, 0.6, 0.7, 0.8, 0.9}.

## D    Additional Results

### D.1    Influence Functions without the Gauss-Newton Hessian Approximation

In all experiments, we computed influence functions using the Gauss-Newton Hessian (GNH) approximation. As the previous error analysis was conducted without the GNH approximation [Basu et al., 2020a], we repeated a subset of our experiments without the GNH approximation (using the Hessian matrix). The results are summarized in Table 4.

As the warm-start gap, proximity gap, non-convergence gap, and linearization error do not depend on the way influence estimates are computed, these numerical values are identical to the results in Table 5. However, as the linearized PBRF optimum is equivalent to influence estimations with the GNH approximation, the solver error slightly increased when we computed influence functions with the Hessian for all datasets. Nevertheless, the solver error is still significantly lower than other decomposition terms.

Furthermore, we investigated how the test loss difference on randomly selected test examples approximated by influence functions correlates with the actual value computed using cold-start retraining, warm-start retraining, and the PBRF. Table 6 shows the correlations with influence approximations using the GNH approximation. While influence estimates do not accurately predict the effect of retraining the model, they closely align the values obtained by the PBRF. Moreover, we conducted the same experiment with influence approximations without the GNH approximation (using the Hessian matrix) and show the results in Table 7. Similar to the previous results, the test loss

| Model | Dataset | Warm-Start | Proximity | Non-Convergence | Linearization | Solver |
|---|---|---|---|---|---|---|
| LR | Cancer | $0.000 \pm 0.000$ | $0.000 \pm 0.000$ | $0.000 \pm 0.000$ | $0.000 \pm 0.000$ | $0.000 \pm 0.000$ |
| | Diabetes | $0.000 \pm 0.000$ | $0.000 \pm 0.000$ | $0.000 \pm 0.000$ | $0.000 \pm 0.000$ | $0.000 \pm 0.000$ |
| MLP | Concrete | $0.079 \pm 0.008$ | $0.002 \pm 0.000$ | $\mathbf{0.082} \pm 0.001$ | $0.000 \pm 0.000$ | $0.001 \pm 0.002$ |
| | Energy | $\mathbf{0.016} \pm 0.001$ | $0.001 \pm 0.000$ | $\mathbf{0.016} \pm 0.001$ | $0.000 \pm 0.000$ | $0.000 \pm 0.000$ |
| | MNIST | $\mathbf{0.208} \pm 0.732$ | $0.006 \pm 0.024$ | $0.211 \pm 0.745$ | $0.000 \pm 0.000$ | $0.001 \pm 0.004$ |
| | FashionMNIST | $\mathbf{2.968} \pm 0.545$ | $0.385 \pm 0.028$ | $0.408 \pm 0.033$ | $0.000 \pm 0.001$ | $0.007 \pm 0.010$ |
| Autoencoder | MNIST | $\mathbf{20.743} \pm 0.522$ | $14.330 \pm 0.184$ | $11.409 \pm 0.430$ | $0.000 \pm 0.000$ | $0.307 \pm 0.088$ |
| LeNet | MNIST | $\mathbf{7.434} \pm 1.068$ | $5.393 \pm 0.547$ | $3.748 \pm 0.324$ | $0.000 \pm 0.001$ | $0.001 \pm 0.001$ |
| AlexNet | | $\mathbf{13.403} \pm 1.289$ | $0.001 \pm 0.000$ | $0.118 \pm 0.000$ | $0.000 \pm 0.000$ | $0.013 \pm 0.002$ |
| VGG13 | | $7.389 \pm 0.744$ | $4.872 \pm 1.052$ | $\mathbf{8.601} \pm 0.478$ | $0.000 \pm 0.000$ | $0.001 \pm 0.001$ |
| ResNet-20 | | $4.433 \pm 0.059$ | $4.061 \pm 0.157$ | $\mathbf{4.940} \pm 0.155$ | $0.001 \pm 0.000$ | $0.002 \pm 0.001$ |
| LeNet | CIFAR10 | $\mathbf{10.668} \pm 0.162$ | $6.520 \pm 0.442$ | $5.032 \pm 0.799$ | $0.001 \pm 0.000$ | $0.000 \pm 0.000$ |
| AlexNet | | $\mathbf{7.530} \pm 0.233$ | $5.956 \pm 0.102$ | $2.864 \pm 0.367$ | $0.000 \pm 0.000$ | $0.000 \pm 0.000$ |
| VGG13 | | $\mathbf{8.410} \pm 1.926$ | $6.279 \pm 0.708$ | $6.031 \pm 0.660$ | $0.000 \pm 0.000$ | $0.073 \pm 0.161$ |
| ResNet-20 | | $\mathbf{5.827} \pm 0.152$ | $4.435 \pm 0.669$ | $3.280 \pm 0.429$ | $0.000 \pm 0.000$ | $0.003 \pm 0.001$ |
| Transformer | PTB | $57.926 \pm 5.055$ | $30.756 \pm 0.673$ | $\mathbf{61.675} \pm 1.042$ | $5.002 \pm 1.556$ | $3.316 \pm 2.630$ |

**Table 5:** Decomposition of the discrepancy between influence functions and LOO retraining into (1) warm-start gap, (2) proximal gap, (3) non-convergence gap, (4) linearization error, and (5) solver error for each model and dataset. The size of each component is measured by the $L_2$ distance between the networks' outputs on the training dataset.

| Dataset | Cold-Start | | Warm-Start | | PBRF | |
|---|---|---|---|---|---|---|
| | P | S | P | S | P | S |
| Concrete | -0.11 | 0.12 | 0.09 | 0.11 | **0.93** | **0.94** |
| Energy | 0.03 | 0.04 | 0.09 | 0.13 | **0.97** | **0.91** |
| MNIST | -0.10 | 0.01 | 0.22 | 0.35 | **0.98** | **0.91** |
| FashionMNIST | 0.16 | 0.00 | 0.06 | 0.07 | **0.90** | **0.92** |

**Table 6:** Comparison of test loss differences computed by influence function (**with** the Gauss-Newton Hessian approximation), cold-start retraining, warm-start retraining, and PBRF. We show Pearson (P) and Spearman rank-order (S) correlation when compared to influence estimates.

differences predicted by influence functions align with the value obtained by the PBRF while failing to capture the effect of retraining the model for both regression and classification datasets. Hence, in both settings, the PBRF better captures the behavior of influence functions than LOO retraining.

## D.2 Two-Stage LOO Retraining: An Alternative Method for PBRF computation

As discussed in Section 4.3, when the network has not fully converged, the LOO retraining simply reflects the effect of training the network for a longer period of time, which does not correctly reflect the effect of removing a data point from the training set (a question that influence functions aim to answer). This difference yields the discrepancy between LOO retraining and influence estimates. We further introduced the PBRF objective which penalizes the LOO retraining with both function- and weight-space discrepancy terms and showed that the PBRF better reflects the question influence functions try to answer.

Here, we discuss an alternative method to capture the behaviour of influence functions by performing two separate LOO retrainings, one with the full dataset and the other one with the removed data point. The first LOO retraining reflects the effect of training longer with the full dataset and the second LOO retraining reflects the effect of training longer with a removed data point. The difference between LOO retrainings in parameter space can approximate the effect of removing a data point by neglecting the effect of longer training. Such difference can be added to the current parameters $\theta^s$. This two-stage LOO retraining method can be interpreted as removing $\mathcal{O}(1)$ terms when the network has not fully converged.

We repeat the experiment from Appendix D.1 with the proposed two-stage LOO retraining. The results are shown in Table 8. While the two-stage LOO retraining shows a worse correlation with

| Dataset | Cold-Start | | Warm-Start | | PBRF | |
|---|---|---|---|---|---|---|
| | P | S | P | S | P | S |
| Concrete | -0.14 | 0.32 | 0.24 | 0.37 | **0.85** | **0.71** |
| Energy | -0.01 | -0.24 | 0.44 | 0.57 | **0.95** | **0.83** |
| MNIST | -0.16 | -0.12 | 0.21 | -0.14 | **0.98** | **0.81** |
| FashionMNIST | -0.19 | 0.01 | 0.05 | 0.24 | **0.91** | **0.88** |

**Table 7:** Comparison of test loss differences computed by influence function (**without** the Gauss-Newton Hessian approximation), cold-start retraining, warm-start retraining, and PBRF. We show Pearson (P) and Spearman rank-order (S) correlation when compared to influence estimates.

| Dataset | Two-Stage LOO | | Warm-Start | | PBRF | |
|---|---|---|---|---|---|---|
| | P | S | P | S | P | S |
| Concrete | 0.52 | 0.61 | 0.24 | 0.37 | **0.85** | **0.71** |
| Energy | 0.92 | 0.75 | 0.44 | 0.57 | **0.95** | **0.83** |
| MNIST | 0.73 | 0.53 | 0.21 | -0.14 | **0.98** | **0.81** |
| FashionMNIST | 0.69 | 0.52 | 0.05 | 0.24 | **0.91** | **0.88** |

**Table 8:** Comparison of test loss differences computed by influence function, warm-start retraining, PBRF, and two-stage LOO retraining on MNIST dataset. We show Pearson (P) and Spearman rank-order (S) correlation when compared to influence estimates.

influence estimates compared to the PBRF objective, it has a significantly higher correlation with influence estimates compared to the warm-start retraining. Although the two-stage LOO retraining requires retraining the network twice, it can be seen as a better reflection to what influence functions compute in neural networks.

### D.3 Mislabelled Examples Detection

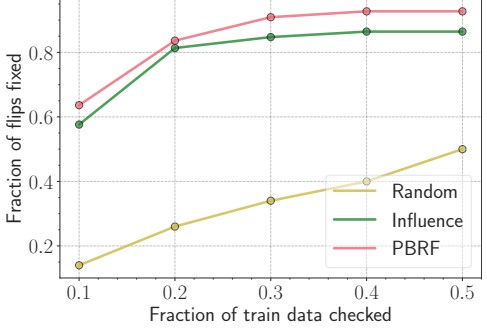

**Figure 6:** Effectiveness of PBRF and influence functions on fixing mislabeled training examples on corrupted MNIST dataset. We examine the fraction of the training data to fix the mislabelling while prioritizing the data examples with higher influential scores produced by PBRF and influence functions. The PBRF and influence functions help detect mislabelled examples.

While the PBRF may not necessarily align with LOO retraining because of warm-start, proximity, and non-convergence gaps, the motivating use cases for influence functions typically do not rely on exact LOO retraining. Hence, the PBRF can be used instead of LOO retraining for many tasks, such as identifying influential or mislabelled examples. We conducted an additional experiment to verify that the PBRF (and influence function) is still helpful in detecting mislabelled training data points.

We used 10% of the MNIST dataset and randomly corrupted 10% of the training examples by assigning random labels. Then, we simulated the scenario where we manually inspect a fraction of training examples, correcting them if they were mislabelled. We trained 2-hidden layer MLP with 1024 hidden units using SGD with a batch size of 128. We used the damping term of $\lambda = 0.001$ for

PBRF and influence functions. For each PBRF and influence estimation, we measured the influence of removing a single training training example on the total training loss (self-influence scores [Koh and Liang, 2017, Khanna et al., 2019]) to identify the top influential training examples.

We prioritized inspecting training examples that obtained high scores generated by the PBRF and influence functions. The results are summarized in Figure 6. Using the self-influence score generated by the PBRF, it is possible to detect over 80% of the mislabelled training examples by only examining 20% of training examples. Similarly, as influence functions closely align with the PBRF, influence functions can provide an efficient tool to help fix mislabelled training examples. Both PBRF and influence functions outperform the baseline of randomly selecting a subset of training examples to inspect if there are any mislabelled training examples.

## E   Influence Misalignment Decomposition Table

In Table 5, we present the numerical results shown in Figure 4.

## G   Efficient `iHVP` Computation

One of the major challenges in applying influence functions to neural networks (Eqn. 4) is that they involve the computation of an inverse-Hessian vector product (`iHVP`). However, for large networks, computing `iHVP`s exactly via storing and inverting the Hessian is intractable. To circumvent this, Koh and Liang [2017] consider alternative methods for approximating `iHVP`s, namely, the method of conjugate gradients (`CG`) [Martens et al., 2010] and the Linear time Stochastic Second-Order Algorithm (`LiSSA`) [Agarwal et al., 2016].

**Conjugate Gradient.**   Given a positive-definite damped Hessian $\nabla_{\boldsymbol{\theta}}^2 \mathcal{J}(\boldsymbol{\theta}^s) + \lambda \mathbf{I}$ (where $\boldsymbol{\theta}^s$ are the potentially-suboptimal parameters at which the Hessian is taken and $\lambda > 0$ is a damping factor) and vector $\mathbf{v} \in \mathbb{R}^d$, `CG` arrives at the `iHVP` by solving an equivalent convex quadratic optimization problem:

$$(\nabla_{\boldsymbol{\theta}}^2 \mathcal{J}(\boldsymbol{\theta}^s) + \lambda \mathbf{I})^{-1} \mathbf{v} = \underset{\mathbf{t} \in \mathbb{R}^d}{\arg\min} \frac{1}{2} \mathbf{t}^\top (\nabla_{\boldsymbol{\theta}}^2 \mathcal{J}(\boldsymbol{\theta}^s) + \lambda \mathbf{I}) \mathbf{t} - \mathbf{v}^\top \mathbf{t}. \tag{47}$$

The `CG` algorithm starts with an initial guess $\mathbf{v}_0 \in \mathbb{R}^d$ and iteratively updates it, with the bottleneck at each step being an $O(Nd)$ Hessian-vector product. Although an exact solution is only guaranteed after $d$ `CG` iterations, in practice, Koh and Liang [2017] use truncated `CG` with fewer iterations and achieve a sufficiently close approximation.

**LiSSA.**   The `LiSSA` algorithm approximates the `iHVP` using a truncated Neumann series. Given a positive-definite damped Hessian $\nabla_{\boldsymbol{\theta}}^2 \mathcal{J}(\boldsymbol{\theta}^s) + \lambda \mathbf{I}$ and vector $\mathbf{v} \in \mathbb{R}^d$, we have:

$$(\nabla_{\boldsymbol{\theta}}^2 \mathcal{J}(\boldsymbol{\theta}^s) + \lambda \mathbf{I})^{-1} \mathbf{v} \approx \sigma^{-1} \sum_{t=1}^{T} ((1 - \sigma^{-1}\lambda)\mathbf{I} - \sigma^{-1} \nabla_{\boldsymbol{\theta}}^2 \mathcal{J}(\boldsymbol{\theta}^s))^t \mathbf{v}, \tag{48}$$

which becomes exact as the recursion depth $T$ approaches $\infty$. Here, $\sigma > 0$ is a scaling hyperparameter that is chosen sufficiently large to ensure convergence of the series. Eqn. 48 can be recursively computed over $T$ iterations, with each step requiring an $O(Nd)$ Hessian-vector product. In practice, the computation is further optimized by estimating $\nabla_{\boldsymbol{\theta}}^2 \mathcal{J}(\boldsymbol{\theta}^s)$ using a randomly-sampled batch $\mathcal{B} \subseteq \mathcal{D}_{\text{train}}$ of size $|\mathcal{B}| \ll N$, so that the Hessian-vector product is reduced to $O(d)$ cost. Then, to accommodate for the added stochasticity, the `iHVP` is estimated by averaging Eqn. 48 over $R$ trial repeats. Hence, the `LiSSA` algorithm estimates:

$$(\nabla_{\boldsymbol{\theta}}^2 \mathcal{J}(\boldsymbol{\theta}^s) + \lambda \mathbf{I})^{-1} \mathbf{v} \approx \frac{1}{R} \sum_{r=1}^{R} \left( \sigma^{-1} \sum_{t=1}^{T} ((1 - \sigma^{-1}\lambda)\mathbf{I} - \sigma^{-1} \nabla_{\boldsymbol{\theta}}^2 \mathcal{J}^{(r,t)}(\boldsymbol{\theta}^s))^t \mathbf{v} \right), \tag{49}$$

where $\mathcal{J}^{(r,t)}(\boldsymbol{\theta}^s)$ is the average loss over the $(r, t)$-th sampled batch of data.

**The $s_{\text{test}}$ trick.**   Finally, we note that another simple trick can be made when using influence functions to predict the change in test loss at a particular test point (Eqn. 20). It is often the case that we wish to compute the influence scores for the pairwise interactions of the entire training dataset on the loss at

a comparatively smaller number $N_{\text{test}} \ll N$ of test points. Since the Hessian is symmetric, the order of multiplication in the second term of Eqn. 20 can be permuted as follows:

$$\frac{1}{N}\mathbf{v}_{\text{test}}^{\top}\left[(\nabla_{\boldsymbol{\theta}}^{2}\mathcal{J}(\boldsymbol{\theta}^{s}) + \lambda\mathbf{I})^{-1}\mathbf{v}\right] = \frac{1}{N}\mathbf{v}^{\top}\left[(\nabla_{\boldsymbol{\theta}}^{2}\mathcal{J}(\boldsymbol{\theta}^{s}) + \lambda\mathbf{I})^{-1}\mathbf{v}_{\text{test}}\right], \tag{50}$$

where $\mathbf{v} = \nabla_{\boldsymbol{\theta}}\mathcal{L}(f(\boldsymbol{\theta}^{s}, \mathbf{x}), \mathbf{t})$ and $\mathbf{v}_{\text{test}} = \nabla_{\boldsymbol{\theta}}\mathcal{L}(f(\boldsymbol{\theta}^{s}, \mathbf{x}_{\text{test}}), \mathbf{t}_{\text{test}})$. This means that we can precompute $\mathbf{s}_{\text{test}} = (\nabla_{\boldsymbol{\theta}}^{2}\mathcal{J}(\boldsymbol{\theta}^{s}) + \lambda\mathbf{I})^{-1}\mathbf{v}_{\text{test}}$ over all $N_{\text{test}}$ test points of interest, and then cheaply compute influence scores over all $N$ training points by simply taking dot products of the form $\mathbf{v}^{\top}\mathbf{s}_{\text{test}}$. We refer readers to Koh and Liang [2017] for details.