# OpenReview forum: "If Influence Functions are the Answer, Then What is the Question?"
_NeurIPS.cc/2022/Conference — NeurIPS 2022 Accept_

### Official Review · Reviewer_br1N · 2022-07-03

**Rating:** 7
**Confidence:** 5
**Soundness:** 4 excellent
**Presentation:** 3 good
**Contribution:** 4 excellent

**Summary:**

 The paper tackles  a very important topic about the evaluation of influence function based methods. Currently, the effectiveness of influence functions are based on (a) Direct metrics such as Leave-out-retraining OR (b) Indirect metrics such as detection of mislabeled samples etc. The authors propose to use proximal Bregman response function as a way to measure the effectiveness or alignment of influence functions, especially for deep models.

**Questions:**

The questions are added in the Weaknesses section.

**Limitations:**

- Lack of motivation on the design of the PBRF metric.

- More analysis is required on Imagenet scale datasets. Although IF is tricky to compute on Imagenet, it would be beneficial to understand the proposed metric at scale.

Overall, I feel the paper is strong and does a well laid out analysis on the alignment of influence estimation and LOO re-training which is lacking in the community. I would also urge the authors to think about in what practical settings and how this metric can be used instead of the commonly used indirect metrics such as mislabeled data points detection.



**Strengths And Weaknesses:**

Strengths:
- The paper does a comprehensive analysis of the different components (warm start, non-convergence, linearization, addition of iHVP etc. ) involved in the misalignment between the influence function and the LOO training objectives. The analysis is interesting and well-executed.

- The authors acknowledge that in principle the deep models are only partially trained as they always don't converge to the optima. Hence, taking this information into account and designing a gold-standard ground-truth is impactful to understand the effectiveness of influence functions.

- The array of experiments are solid and cover a wide range of architectures and datasets / domains. The paper is a good extension to Basu et al (ICLR 2021)[1]  and follows up that paper with more in-depth experiments.

Weaknesses:
- While the PBRF metric aligns well with the influence score, I would like to see some more motivation about why the authors thought it’s a good metric to evaluate influence functions on.  The current version of the paper lacks this motivation or information corresponding to it.

- If we already know IFs are good at other metrics such as detection of mislabeled examples / relabeling, why do we require a fix to the evaluation procedure of influence functions. It would be good if the authors could substantiate on why alignment of IF with a gold standard ground-truth is important as opposed to testing them on other indirect but more practical metrics such as detection of mislabeled examples / relabeling.

- While the authors cover a wide range of experiments, I would like to see some experiments on more complex datasets such as Imagenet-1k. A focused analysis on Imagenet-1k would make the paper stronger.


[1]. https://arxiv.org/abs/2006.14651?context=stat.ML

---

> ### Author Response · Authors · 2022-08-02
> **Author Response**
>
> Thank you for your feedback and help to improve the paper! We are excited that the reviewer found our paper to be strong.
>
> > **Q: “While the PBRF metric aligns well with the influence score, I would like to see some more motivation about why the authors thought it’s a good metric to evaluate influence functions on. The current version of the paper lacks this motivation or information corresponding to it.”**
>
> Appendix B.3 provides a more in-depth analysis of the PBRF. Specifically, we show that influence functions evaluated at non-converged parameters are equivalent to the first-order approximation of the PBRF for linear models. Note that this is not true for the proximal response function (LOO retraining with and without damping); when the cost gradient is non-zero, influence functions are not equivalent to the first-order approximation of the proximal response function even for linear models (please see Section 4-3 for more details). We made this argument more explicit in the updated manuscript.
>
> In addition to our theoretical motivation from linear models, the PBRF also has an interesting interpretation where it computes the training error replaced with the soft targets obtained from the non-converged parameters as detailed in Section 4-3.
>
> > **Q: “It would be good if the authors could substantiate on why alignment of IF with a gold standard ground-truth is important as opposed to testing them on other indirect but more practical metrics such as detection of mislabeled examples / relabeling.”**
>
> The effectiveness of influence-based methods in neural networks is generally evaluated by (1) observing an alignment with LOO retraining (e.g., [1]) or (2) proxy metrics such as the recovery of maliciously corrupted examples using influence scores (e.g., [2]). In this paper, we show that LOO retraining is an inadequate metric as it does not correctly reflect what influence functions approximate.
>
> In cases where the downstream task is to detect mislabeled examples / relabeling, we believe considering these proxy metrics is helpful. However, for the general purpose of evaluating influence functions, we believe that our framework provides a more direct signal in developing algorithmic improvements. If one is interested in improving certain aspects of influence function estimation, such as the linear system solver (e.g., more accurate computation of iHVP), it is important to have a well-defined quantity that influence functions more closely approximate so that algorithmic choices can be directly evaluated based on the accuracy of their estimates.
>
> To give a more concrete example, let us say one developed a linear solver for influence functions. One evaluated the solver on one of these proxy metrics and obtained better results. However, it is still unclear if the improvements are due to more accurate iHVP computation or some other factors such as inductive bias. In contrast, evaluating the algorithm on our framework provides a more direct signal in what aspects of the new linear solver improved.
>
> > **Q: “While the authors cover a wide range of experiments, I would like to see some experiments on more complex datasets such as Imagenet-1k. A focused analysis on Imagenet-1k would make the paper stronger.”**
>
> We agree with the reviewer that larger-scale experiments such as ImageNet-1k would strengthen our paper. However, as detailed in Appendix C.2, each analysis requires 100 retrainings of the model, 20 iHVP evaluations, and additional trials to tune the hyperparameters. Repeating our framework on larger-scale experiments would require significant computational resources.
>
> [1] Basu, S., Pope, P., & Feizi, S. (2020). Influence functions in deep learning are fragile. arXiv preprint arXiv:2006.14651.
>
> [2] Khanna, R., Kim, B., Ghosh, J., & Koyejo, S. (2019, April). Interpreting black box predictions using fisher kernels. In The 22nd International Conference on Artificial Intelligence and Statistics (pp. 3382-3390). PMLR.

---

> > ### Comment · Reviewer_br1N · 2022-08-08
> > **Response to Authors**
> >
> > Thank you for the rebuttal response.
> >
> > I feel this work solves a major problem in evaluating influence functions, hence I would recommend a strong acceptance.
> >
> > To make this work more impactful, I would still urge to (a) Include experiments on more real-world datasets to validate the proposed metric; (b) a note on the usefulness of the metric (i.e., why a gold standard is required) in the final version of the paper.

---

### Official Review · Reviewer_TgaB · 2022-07-10

**Rating:** 6
**Confidence:** 2
**Soundness:** 3 good
**Presentation:** 3 good
**Contribution:** 3 good

**Summary:**

The paper studies what causes influence estimates misalign the leave-one-out retraining. Specifically, the authors decompose discrepancy into five separate terms and empirically show the contributions of each term under a variety of settings (e.g., model architectures and datasets).

**Questions:**

1. What is the purpose of the deviation from Eq. 7 to Eq. 9?
2. In eq. 8, the authors mention “$f_{\text{lin}}(\cdot, \cdot)$ is the linearization of the network outputs with respect to the parameters”.
3. What do you mean by “the linearization of the network outputs”? Is there any practical advantage of using over PBRF influence estimates?

**Limitations:**

The authors have adequately addressed the limitations and potential negative societal impact of their work.

**Strengths And Weaknesses:**

*Originality*: The paper shows the connections between influence estimates and the decomposition of the five terms, and finds out that influence estimates are highly correlated to the proposed proximal Bregman response function (PBRF). The results are overall interesting.

*Quality*: The decomposition is technically sound and empirical analysis of the decomposition is well conducted. However, I suggest the authors show some use cases of PBRF empirically. As demonstrated in the experiment, PBRF is highly correlated with influence estimates. Does it offer some better advantage over influence estimates in some use cases such as data poisoning, improving fairness and explanablity of the model. I think these are also key questions to answer.

*Clarity*: The paper is well written and easy to read. However, some technical details of the paper are not well explained. For example, I am a little confused by the introduction of Linearization Error (from Eq. 7 to Eq. 9). I am not sure what are the purposes/usages by performing the second-order expansion of the loss around $y_s$.

*Significance*: This paper provides an in-depth understanding of influence function by trying to bridge the gap between influence function and LOO re-training . It could impact on the community in the long run.

---

> ### Author Response · Authors · 2022-08-02
> **Author Response**
>
> Thank you for your feedback and help to improve the paper!
>
> > **Q: What are the use cases of PBRF? “Does it offer some better advantage over influence estimates in some use cases such as data poisoning, improving fairness and explanablity of the model?”**
>
> As the reviewer pointed out, one use case of the PBRF is for a more accurate estimation of the effect of removing a data point, which can potentially help in various applications involving influence functions (e.g., improving data poisoning and fairness). In Appendix D.2, we use the PBRF and influence functions to detect mislabelled training images. As the PBRF eliminates (4) linearization and (5) solver errors, it better detects mislabelled training images than influence functions, as expected. We added a direct reference to Appendix D.2 in our manuscript.
>
> The PBRF also helps in strengthening the understanding of influence functions. For example, [1] claimed that influence functions in neural networks are often erroneous due to the misalignment between LOO retraining and influence estimates. However, we show that influence functions are not erroneous but simply answer a different question than anticipated. We believe our formulation yields valuable insights into what influence functions compute in neural networks.
>
> Finally, we can use our decomposition for validation and algorithm research, as it provides a well-defined quantity that influence functions are meant to approximate. For example, if one is interested in improving certain aspects of influence function estimation, such as the linear system solver, it would be preferable to have a well-defined quantity that influence function estimators are approximating so that algorithmic choices could be directly evaluated based on the accuracy of their estimates.
>
> > **Q: Clarification of the linearization error; “The purpose of the deviation from Eq. 7 to Eq. 9.”**
>
> The optimal solution to the linearized PBRF (Eq. 9) is equivalent to the influence estimation with the GNH approximation. We present the derivation in Appendix B.4, including the proper formulation of $\mathcal{L}{\text{quad}}$ and $f{\text{lin}}$. The purpose of linearization and quadratic approximation in Eq. 9 is to reflect the approximations made in the first-order influence function. We apologize for the confusion and clarified this connection in our updated manuscript.
>
> Recall that first-order influence functions approximate the effect of removing a training example by performing a second-order Taylor expansion. By comparing the discrepancy between the PBRF (Eq. 7) and linearized PBRF (Eq. 9) (linearization error), we can analyze the error caused by omitting the higher-order terms in the first-order influence function. For example, in Figure 5 (f), we can observe that the linearization error increases as the number of removed data increases; the local approximation that influence functions rely on becomes more inaccurate as the perturbation in the dataset increases.
>
> > **Q: “What do you mean by the linearization of the network output? Is there any practical advantage of using over PBRF influence estimates?”**
>
> We present the precise definition in Appendix B.4. We also added it in Appendix A (Table 3) in our updated manuscript for clarification. We linearized the network outputs so that the optimal solution to linearized PBRF is equivalent to the influence estimation in Eqn. 5. Note that the Gauss-Newton Hessian (GNH) approximation can be understood as the linearization of the network’s output [2]. We made this connection explicit in the updated manuscript.
>
> Please see the reply to the previous question about why linearized PBRF is an important objective to analyze. To give a more concrete example, one can consider using higher-order influence functions [3] for some practical applications when the discrepancy between the PBRF and linearized PBRF is large.
>
> [1] Basu, S., Pope, P., & Feizi, S. (2020). Influence functions in deep learning are fragile. arXiv preprint arXiv:2006.14651.
>
> [2] Martens, J. (2020). New insights and perspectives on the natural gradient method. The Journal of Machine Learning Research, 21(1), 5776-5851.
>
> [3] Koh, P. W. W., Ang, K. S., Teo, H., & Liang, P. S. (2019). On the accuracy of influence functions for measuring group effects. Advances in neural information processing systems, 32.

---

> > ### Comment · Reviewer_TgaB · 2022-08-09
> > **Response to Authors**
> >
> > Thanks for your response! My concerns have been addressed.

---

### Official Review · Reviewer_uyV1 · 2022-07-11

**Rating:** 7
**Confidence:** 4
**Soundness:** 3 good
**Presentation:** 3 good
**Contribution:** 3 good

**Summary:**

This paper studies influence functions. Influence functions work well with leave-one-out retraining for linear models, but they often have poor performance in neural networks. The authors investigate the specific factors that cause this discrepancy and decompose into five terms. The authors then study the contributions of each term. Furthermore, the authors show that influence functions are often a good approximation to the proximal Bregman response function (PBRF). The experimental results suggest that current algorithms for influence function estimation can still give more informative results than previous error analyses would suggest. In my opinion, the main contribution of this work is a more in-depth analysis of influence functions, especially as it applies to neural networks. Overall, the paper is well organized and well written.

**Questions:**

Here are some questions for the authors:
- As mentioned in weaknesses, what is the underlying reason for selecting the 5 components?
- Are there any considerations in the selection of experimental datasets?


**Limitations:**

The authors adequately addressed the limitations.

**Strengths And Weaknesses:**

Strengths:
- This work investigates the source of the discrepancy between influence functions and LOO retraining in neural networks, and further decompose the discrepancy into five detailed components.
- The authors evaluate the contributions of each component on binary classification, regression, image classification, and language modeling tasks.
- The experimental results show that influence functions are a much better match to the proximal Bregman response function (PBRF), and also can give more informative results than previous error analyses would suggest.

Weaknesses:
- What is the underlying reason for selecting the 5 components and why are these components considered important? These are not explicitly stated in the paper.
- The author considers PBRF can be regarded as the gold standard, and I think more rigorous proofs are needed.

---

> ### Author Response · Authors · 2022-08-02
> **Author Response**
>
> Thank you for your feedback and help to improve the paper!
>
> > **Q: “What is the underlying reason for selecting the 5 components and why are these components considered important?”**
>
> The five components in our analysis are designed to capture all approximations made or assumptions violated when deploying influence functions in neural networks.
> 1. The warm-start gap measures the discrepancy between warm-start and cold-start retraining, and this gap is greater than 0 when the objective is not strongly convex. *[assumption violated]*
> 2. The proximity gap measures the discrepancy between warm-start retraining and proximal warm-start retraining, and the gap is greater than 0 when a damping term is included, a common practice for numerical stability and ensuring positive definiteness of the Hessian [3]. *[assumption violated, approximation]*
> 3. The non-convergence gap measures the discrepancy between proximal warm-start retraining and Proximal Bregman Response Function (PBRF). This gap is higher than 0 when the influence function is computed on non-converged parameters. *[assumption violated]*
> 4. Lastly, linearization error reflects the approximation error caused by second-order Taylor expansion in the influence function derivation *[approximation]*, and solver error represents the approximation error caused by using approximate linear solvers (e.g., truncated CG and Lissa). *[approximation]*
>
> By decomposing the discrepancy into these five terms, we can analyze how each violation in assumption (e.g., objective not being strongly convex) or approximation (e.g., using second-order Taylor expansion) contributes to the total discrepancy between LOO retraining and influence estimates, as done in Section 6.2. We updated our manuscript to clarify this point further.
>
> > **Q: “The author considers PBRF can be regarded as the gold standard, and I think more rigorous proofs are needed.”**
>
> Appendix B.3 provides a more in-depth analysis of the PBRF. Specifically, we show that influence functions evaluated at non-converged parameters are equivalent to the first-order approximation of the PBRF for linear models. Note that this is not true for the proximal response function (LOO retraining with and without damping); when the cost gradient is non-zero, influence functions are not equivalent to the first-order approximation of the proximal response function even for linear models (see Section 4-3 for more details). We made this argument more explicit in the updated manuscript.
>
> > **Q: “Are there any considerations in the selection of experimental datasets?”**
>
> We utilized the Cancer and Diabetes dataset following a logistic regression experiment from [1]. In image classification experiments, we used 10% of MNIST data and the entire CIFAR10 dataset following [2] and [3]. We also used the FashionMNIST to increase diversity in the dataset. We used Concrete and Energy datasets, which are commonly used in regression experiments (e.g., [4]). Finally, we utilized the PTB dataset, which is also a commonly used benchmark in language modeling experiments (e.g., [5]).
>
> [1] Kong, S., Shen, Y., & Huang, L. (2021, September). Resolving Training Biases via Influence-based Data Relabeling. In International Conference on Learning Representations.
>
> [2] Basu, S., Pope, P., & Feizi, S. (2020). Influence functions in deep learning are fragile. arXiv preprint arXiv:2006.14651.
>
> [3] Koh, P. W., & Liang, P. (2017, July). Understanding black-box predictions via influence functions. In International conference on machine learning (pp. 1885-1894). PMLR.
>
> [4] Izmailov, P., Vikram, S., Hoffman, M. D., & Wilson, A. G. G. (2021, July). What are Bayesian neural network posteriors really like?. In International conference on machine learning (pp. 4629-4640). PMLR.
>
> [5] Gal, Y., & Ghahramani, Z. (2016). A theoretically grounded application of dropout in recurrent neural networks. Advances in neural information processing systems, 29.

---

### Official Review · Reviewer_VBq7 · 2022-07-13

**Rating:** 6
**Confidence:** 4
**Soundness:** 3 good
**Presentation:** 2 fair
**Contribution:** 2 fair

**Summary:**

The influence function-based method can analyze the effects of adding/removing/perturbating a single data point on parametric models using a linear approximation at the optimal point of parameters. It was pointed out that this approximation on deep neural networks is poor in previous works.

In this paper, the authors push it further and analyze the factors that affect the approximation, and argue that the performance of influence function methods is not necessarily reflected in the LOO re-training process, but rather can be measured using proximal Bregman Response Function. In the experiments, the authors thoroughly analyzed how different factors affect the discrepancy between influence functions and LOO retraining and showed that PBRF better aligns with influence function methods.

**Questions:**

see above

**Limitations:**

authors briefly mentioned that in Q/A

**Strengths And Weaknesses:**

Strengths:
+ The authors perform analysis and make hypotheses on an important problem (the discrepancy between influence function and LOO), and point out five sources that affect the misalignment
+ The authors pointed out that PBRF better captures the behaviour of influence functions
+ The experiment evaluations are thorough

Weaknesses:
- Although identifying the sources that lead to the misalignment is quite interesting, I'm not very convinced that introducing a different score to solely analyze the influence functions is quite useful or necessary. Influence function methods are initially proposed as an interpretability tool to analyze the model behaviour of removing data points. Introduction and emphasizing on PBRF seem to have lost this goal and are leading to a less meaningful direction.
- In general, I think the structure and writing of this paper need to be improved.

---

> ### Author Response · Authors · 2022-08-02
> **Author Response**
>
> Thank you for your feedback and help in improving the paper!
>
> > **Q: “I'm not very convinced that introducing a different score to solely analyze the influence functions is quite useful or necessary.”**
>
> We believe that our analysis is valuable in understanding influence functions. As a motivating example, [1] concluded that influence functions in neural networks are often erroneous due to the misalignment between LOO retraining and influence estimates. On the contrary, other lines of work (e.g., [2, 3]) have demonstrated that influence-based methods are often successful in detecting malicious or mislabeled training examples. Our analysis bridges these gaps and shows that influence functions in neural networks are still helpful but simply answer a different question than anticipated.
>
> Our framework can further yield helpful insights in validation and algorithm research. For example, if one is interested in developing higher-order influence functions, a comparison to linearization error (Section 4.3) would provide a more direct signal and help understand how one's algorithm improves certain aspects of the influence function performance.
>
> > **Q: “Influence function methods are initially proposed as an interpretability tool to analyze the model behaviour of removing data points. Introduction and emphasizing PBRF seem to have lost this goal and are leading to a less meaningful direction.”**
>
> In contrast to the previous analyses [1], our work shows that influence functions are still valuable as an interpretability tool for analyzing model behaviour in neural networks (e.g., detecting mislabeled examples). We believe that our analysis (including PBRF) does not lead to a less meaningful direction but strengthens our understanding of *how* influence functions help in analyzing the model behaviour of removing data points.
>
> > **Q: “In general, I think the structure and writing of this paper need to be improved.”**
>
> Thank you for your comment. In accordance with reviewers TgaB's and br1N's feedback, we have modified the manuscript to explain the motivation for the linearized PBRF objective more clearly and referenced the materials in the Appendix more explicitly.
>
>
> [1] Basu, S., Pope, P., & Feizi, S. (2020). Influence functions in deep learning are fragile. arXiv preprint arXiv:2006.14651.
>
> [2] Schioppa, A., Zablotskaia, P., Vilar, D., & Sokolov, A. (2022, June). Scaling up influence functions. In Proceedings of the AAAI Conference on Artificial Intelligence (Vol. 36, No. 8, pp. 8179-8186).
>
> [3] Kong, S., Shen, Y., & Huang, L. (2021, September). Resolving Training Biases via Influence-based Data Relabeling. In International Conference on Learning Representations.

---

### Author Response · Authors · 2022-08-02
**General Author Response**

We thank all reviewers for their thoughtful reviews and helpful comments! We are pleased that the reviewers identified our work to be well-written and easy to read (uyV1, TgaB), interesting (VBq7, TgaB, br1N), technically solid (VBq7, uyV1, TgaB, br1N), and the experiments to be comprehensive (VBq7, uyV1, TgaB, br1N). No reviewer found any factual mistakes in our paper.

We further address the remaining concerns and questions of all reviewers in individual responses; we have also revised the manuscript to address the reviewers’ questions and comments, with the updates highlighted in blue text. We thank the reviewers again for their time.

---

### Meta-Review · Area_Chair_Dwo2 · 2022-08-26

**Recommendation:** Accept
**Confidence:** Certain

**Metareview:**

All reviewers felt that this paper deserves to be accepted given its interesting analysis of an important problem and compelling empirical evaluations. The authors should consider incorporating the reviewers' comments, particularly around motivating the PBRF as a "gold standard" and clarifying aspects of the presentation, in the final version of the paper.

**Award:**

No

---

### Decision · Program_Chairs · 2022-09-14

Accept